# MUIRBENCH: A COMPREHENSIVE BENCHMARK FOR ROBUST MULTI-IMAGE UNDERSTANDING

**Fei Wang**[1*]  **Xingyu Fu**[2*]  **James Y. Huang**[1†]  **Zekun Li**[3†]  **Qin Liu**[4†]  **Xiaogeng Liu**[5†]
**Mingyu Derek Ma**[6†]  **Nan Xu**[1†]  **Wenxuan Zhou**[1†]  **Kai Zhang**[7]  **Tianyi Yan**[1]  **Wenjie Mo**[4]
**Hsiang-Hui Liu**[3]  **Pan Lu**[6]  **Chunyuan Li**[8]  **Chaowei Xiao**[5]  **Kai-Wei Chang**[6]  **Dan Roth**[2]
**Sheng Zhang**[8]  **Hoifung Poon**[8]  **Muhao Chen**[4]

[1]USC  [2]UPenn  [3]UMN  [4]UC Davis  [5]UW–Madison  [6]UCLA  [7]OSU  [8]Microsoft Research

Project page: `https://muirbench.github.io/`

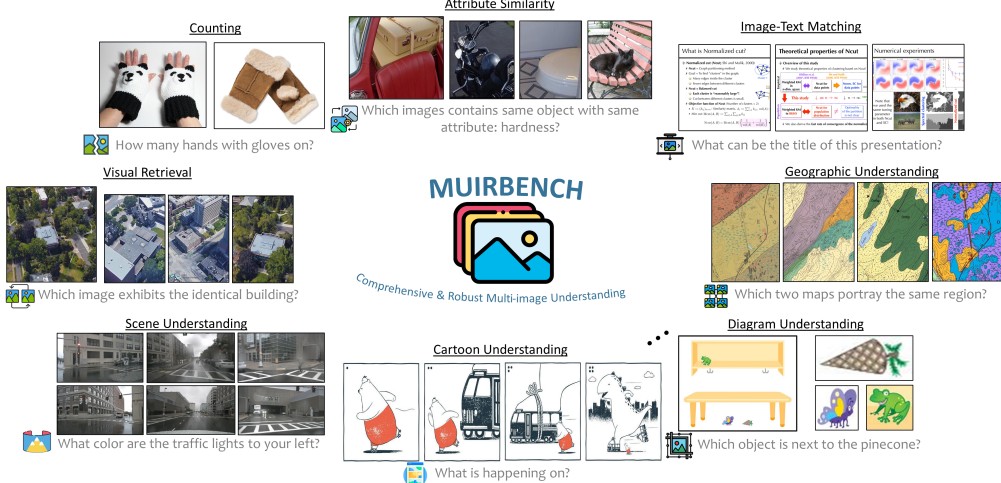

Figure 1: **The MUIRBENCH Benchmark.** MUIRBENCH contains 11,264 images and 2,600 multiple-choice questions, providing robust evaluation on 12 multi-image understanding tasks. Each example comes from one task in MUIRBENCH, presenting diverse multi-image relations.

## ABSTRACT

We introduce MUIRBENCH, a comprehensive benchmark that focuses on robust multi-image understanding capabilities of multimodal LLMs. MUIRBENCH consists of 12 diverse multi-image tasks (*e.g.*, scene understanding, ordering) that involve 10 categories of multi-image relations (*e.g.*, multiview, temporal relations). Comprising 11,264 images and 2,600 multiple-choice questions, MUIRBENCH is created in a pairwise manner, where each standard instance is paired with an unanswerable variant that has minimal semantic differences, in order for a reliable assessment. Evaluated upon 20 recent multi-modal LLMs, our results reveal that even the best-performing models like GPT-4o and Gemini Pro find it challenging to solve MUIRBENCH, achieving 68.0% and 49.3% in accuracy. Open-source multimodal LLMs trained on single images can hardly generalize to multi-image questions, hovering below 33.3% in accuracy. These results highlight the importance of MUIRBENCH in encouraging the community to develop multimodal LLMs that can look beyond a single image, suggesting potential pathways for future improvements.

---

*Equal leadership. Correspondance to <fwang598@usc.edu; xingyuf2@seas.upenn.edu>.
†Equal contribution; alphabetic order.

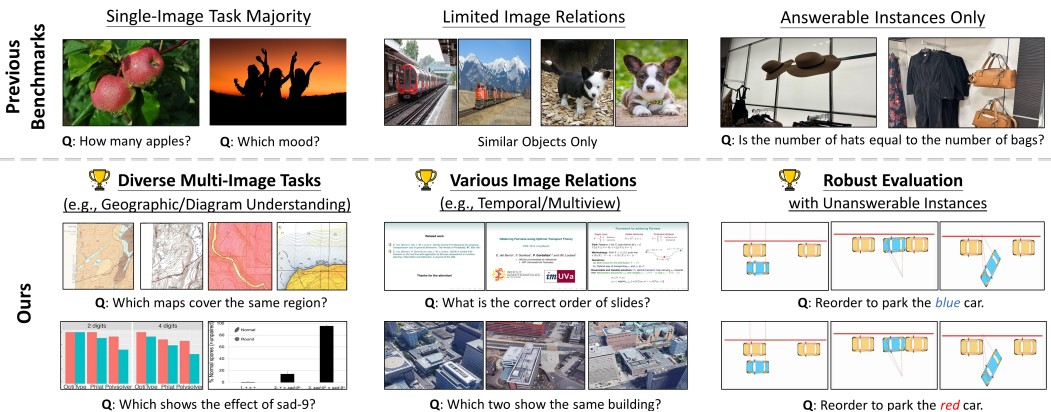

Figure 2: Compared with previous benchmarks, MUIRBENCH has several novel features: (1) It evaluates on a comprehensive range of 12 multi-image understanding abilities, *e.g.* geographic understanding and diagram understanding as introduced in §3, while prior benchmarks generally contain single-image questions. (2) It contains 10 diverse multi-image relations, *e.g.* narrative and complementary as discussed in §3. (3) It provides a robust evaluation on models by introducing unanswerable instance variants. The samples of previous benchmarks are from (Liu et al., 2023c; Suhr et al., 2019; Jiang et al., 2024a).

# 1 INTRODUCTION

The proverb "a picture is worth a thousand words" is often cited to emphasize the richness of visual information hidden in one image (Gropper, 1963; Hibbing & Rankin-Erickson, 2003). However, an image is only a single projection of the real world captured from a specific angle at a specific moment in time (Hays & Efros, 2008). In contrast, humans naturally observe multiple images – multiple pieces of such projections from discrete moments under various scenes – to perceive and understand the world as a holistic part. Humans excel at synthesizing information from multiple image sources, whether it involves telling stories from a series of cartoon images (Cohn et al., 2017; Li et al., 2023), drawing comparisons among multiple charts and diagrams to infer holistic new insights (Masry et al., 2022), learning from diverse visual experiences such as online lesson slides to adopt new skills (Nouri & Shahid, 2005), predicting future event actions from past screenshots (Oh et al., 2015; Finn et al., 2016), or conducting temporal reasoning based on nuanced differences between photographs (Fu et al., 2022). Moreover, multi-image input has the advantage of conveying visuospatial ideas directly – combining multiple images of the same scene can reveal spatial relations or other more abstract relations in the world (Faugeras et al., 2001). Multi-image input also overcomes the limitations of resolution that single-image input faces, allowing for better visual perception and understanding (Kawulok et al., 2019).

As multimodal large language models (LLMs) (OpenAI, 2023; Team et al., 2023; Liu et al., 2023b; 2024a; Alayrac et al., 2022; Bai et al., 2023; Wang et al., 2023; Dai et al., 2023; Lu et al., 2023b; Zhang et al., 2024b; Team, 2024; Chen et al., 2023c; Chaves et al., 2024) have begun to show superior performance across various single-image tasks, we now expect them to solve hard tasks that require an holistic understanding of multiple images. This work aims at highlighting crucial aspects of multi-image understanding that have been overlooked when evaluating multimodal LLMs, and providing a comprehensive benchmark for robust multi-image reasoning. As shown in Figure 2, current evaluations (Goyal et al., 2017; Liu et al., 2023c; Li et al., 2023; Yue et al., 2023; Lu et al., 2024; Liu et al., 2023a;d) generally focus on single-image understanding, thereby neglecting the richer, more complex tasks of integrating and reasoning across multiple images. While many of these benchmarks have been popularized as the de facto evaluation measures for influential models like GPT-4-Turbo (OpenAI, 2023) and Gemini-Pro (Team et al., 2023), this oversight limits the potential of these models to conduct advanced-level multimodal comprehension. Though some recent benchmarks start to include multi-image questions in evaluation (*e.g.*, Mantis-Eval (Jiang et al., 2024a) and BLINK (Fu et al., 2024)), they are far from being comprehensive in multi-image evaluation that involve multi-perspectives, multi-relations and robustness concerns.

In this paper, **we introduce MUIRBENCH (MULTI-IMAGE UNDERSTANDING BENCHMARK)**, a comprehensive benchmark designed to rigorously assess and evaluate multi-image understanding by multimodal LLMs. MUIRBENCH encompasses 11,264 images and 2,600 multiple-choice questions spanning across 12 distinctive multi-image understanding tasks, *e.g.*visual retrieval, cartoon understanding, and attribute similarity, *etc.*. As illustrated in Figure 1, there can be multiple images interleaved in the contexts or questions, or presented as choices in our benchmark. Instances in MUIRBENCH also contain diverse kinds of multi-image relations, *e.g.* temporal, ordered-pages, or narrative relations, *etc*. as shown in Figure 4. The questions and choices are either derived from the datasets, or manually written by experts. Additionally, MUIRBENCH adopts a pairwise design approach, where each question-answering instance is paired with a expert-annotated unanswerable counterpart (Rajpurkar et al., 2018) featuring minimal differences following Figure 5. This design ensures a reliable assessment of multimodal LLMs, mitigating the risk of achieving correct answers through vision or language shortcuts. We also include various fine-grained expert annotated labels such as image positions and image types in MUIRBENCH, to facilitate detailed model analysis.

We conduct a comprehensive evaluation on MUIRBENCH using 20 multimodal LLMs of various sizes, including models that accept multi-image inputs and those originally designed for single-image inputs. Experimental results underscore the current limitations of even the most influential multimodal LLMs, *e.g*. GPT-4o and Gemini Pro, in handling multi-image scenarios. For instance, GPT-4o and Gemini Pro achieve mere 68.0% and 49.3% of accuracy respectively, which are 25.1 % and 43.8% lower than human performance. We also show that multimodal LLMs perform much worse on unanswerable questions than their answerable counterparts, with GPT-4o and Gemini Pro exhibiting accuracy gaps of 26.8% and 21.5%. Furthermore, multimodal LLMs trained solely on single images demonstrate impaired generalization to multi-image contexts. These findings highlight the significance of MUIRBENCH in driving the development of multimodal LLMs in transcending single-image limitations. We believe MUIRBENCH can serve as an effective testbed for holistic multi-image understanding, encouraging the community to cultivate models with a more comprehensive and integrated understanding of the visual world.

## 2 RELATED WORK

### 2.1 MULTIMODAL UNDERSTANDING BENCHMARKS

A number of recent benchmarks have been developed to comprehensively assess the multimodal understanding and reasoning capabilities of multimodal language models (LLMs) (Lu et al., 2021a; 2022; Li et al., 2023; Liu et al., 2023c; Lu et al., 2024; Yue et al., 2023; Zhang et al., 2024c; Ying et al., 2024; Wu & Xie, 2023). However, most of these benchmarks primarily focus on single-image scenarios. While some benchmarks include multi-image examples (Lu et al., 2024; Yue et al., 2023; Fu et al., 2024; Ying et al., 2024; Wang et al., 2024b; Zhao et al., 2024), they typically require limited aspects of capacities (*e.g*., image comparison for MathVista) and do not provide a comprehensive assessment of multimodal LLMs in multi-image scenarios. While some benchmarks feature video understanding (Grauman et al., 2022; Maaz et al., 2023) or in-context learning (Shukor et al., 2023; Jiang et al., 2024b), the assessed capabilities are fundamentally different from multi-image understanding. Video understanding focuses on continuous streams of frames capturing dynamic changes over time, while in-context learning focuses on task adaptation using few-shot examples. In contrast, multi-image understanding challenges models to integrate and analyze spatial and contextual cues from varied perspectives, settings, and moments, thereby simulating the way humans process information from multiple visual sources. Recently, there have been dedicated efforts to assess multimodal LLMs in multi-image scenarios. For example, MANTIS-Eval (Jiang et al., 2024a) is a human-annotated benchmark comprising 207 examples for multi-image reasoning, such as size perceptions and weight comparisons. DEMON (Li et al., 2024c) evaluates whether multimodal LLMs can follow zero-shot demonstrative instructions. MileBench (Song et al., 2024) assesses multimodal LLMs' performance under long-context scenarios. However, these benchmarks either still focus on limited multi-image relations and reasoning processes or lack of controlled and robust evaluation. In contrast, MUIRBENCH provides a comprehensive assessment of multimodal LLMs, covering a broader range of multi-image capacities. We provide a detailed comparison between MUIRBENCH and related benchmarks in Appendix §B.

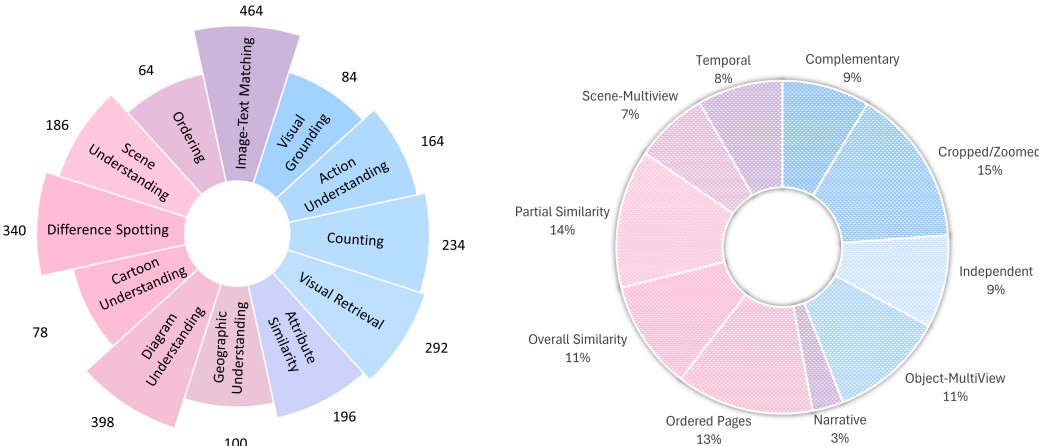

Figure 3: Data distribution by tasks in MUIRBENCH. More details are in §3.

Figure 4: Data distribution by multi-image relation categories. More details are in §3.

## 2.2 MULTIMODAL LARGE LANGUAGE MODELS

Inspired by the remarkable achievements in recent LLMs (Brown et al., 2020; OpenAI, 2023; Touvron et al., 2023; Zheng et al., 2024; Team, 2023b), a series of studies have begun exploring multimodal LLMs that can concurrently interpret visual and linguistic information. However, most of early multimodal LLMs are trained on single-image datasets and overlook the complicated tasks of multi-image understanding (Liu et al., 2023b; Dai et al., 2023; Chen et al., 2023b; Wang et al., 2023), although some of them process each image as multiple tiles (Liu et al., 2024a; Chen et al., 2024a). Recent work starts training multimodal LLMs on interleaved image-text corpus such as MMC4 (Zhu et al., 2024) and OBELICS (Laurençon et al., 2024) for pretraining as well as Mantis-Instruct (Jiang et al., 2024a) for instruction tuning, which enables models to generate texts given multiple images. While some of these models, like Flamingo (Alayrac et al., 2022), Idefics (Laurençon et al., 2024), Emu (Sun et al., 2023a), and VILA (Lin et al., 2023), have demonstrated in-context learning capabilities, there is still a lack of evidence regarding their capabilities in understanding multiple images within independent instances. Although instruction tuned models such as Mantis (Jiang et al., 2024a) and GPT-4-Turbo (OpenAI, 2023) have shown to possess counting and comparison skills over multi-image inputs, their ability in understanding and reasoning over multiple images with different relations across diverse tasks, though critical, remain unexplored. Therefore, we propose MUIRBENCH to conduct comprehensive evaluation and provide insights to further improve their capabilities in handling realistic multi-image tasks.

## 3 MUIRBENCH

Our benchmark is meticulously curated for comprehensively assessing multimodal LLMs' capabilities in holistic multi-image understanding. We introduce the overall design and key features of MUIRBENCH in §3.1, and delve deep into the data curation process in §3.2.

### 3.1 BENCHMARK OVERVIEW

Focusing on multi-image understanding, MUIRBENCH consists of 11,264 images and 2,600 multiple-choice questions, with an average of 4.3 images per instance. In general, MUIRBENCH adheres to two key design principles. First, it seeks to provide a **comprehensive** and holistic evaluation on multimodal LLMs' multi-image understanding capabilities, by containing 12 diverse multi-image tasks covering 10 distinctive multi-image relation categories. Additional fine-grained labels such as input image positions and image types are also included to support comprehensive analysis of models. Second, it seeks to provide a **robust** evaluation, following a pairwise design where each answerable instance is paired with an unanswerable counterpart featuring minimal differences.

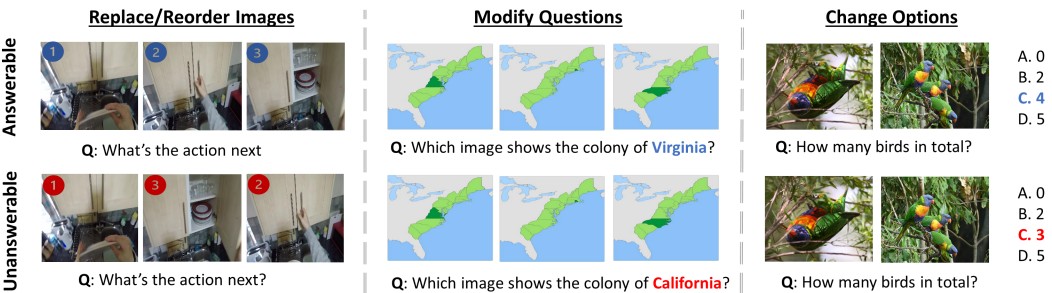

Figure 5: Three major strategies are used in MUIRBENCH to create unanswerable instances from their answerable counterparts with minimal changes (§3.2). In the above examples, blue marks denote the original input in the answerable case, and red marks highlight the input in the unanswerable case.

**Comprehensive Multi-Image Evaluation.** MUIRBENCH provides an comprehensive assessment through 12 distinctive multi-image understanding tasks, with selected examples of each task shown in Figure 6. As illustrated in Figure 3, each task represents 2.5% to 17.8% of the whole benchmark.

[ACTION UNDERSTANDING] aims to evaluate the ability of models to understand continuous images in chronological order and match it with an action. [ATTRIBUTE SIMILARITY] aims to evaluate the ability of models to identify a specific given attribute among multiple images. [CARTOON UNDERSTANDING] aims to evaluate the ability of models to understand stories conveyed in cartoon images. [COUNTING] aims to evaluate the ability of models to count the number of specific objects across multiple images. [DIAGRAM UNDERSTANDING] aims to evaluate the ability of models to understand information conveyed in diagram images. [DIFFERENCE SPOTTING] aims to evaluate the ability of models to identify differences across multiple images. [GEOGRAPHIC UNDERSTANDING] aims to evaluate the ability of models to understand maps and reason upon geographic features. [IMAGE-TEXT MATCHING] aims to evaluate the ability of models to understand the meaning of a text snippet and match it with the corresponding visual content or vice versa. [ORDERING] aims to evaluate the ability of models to order a series of images based on the textual description. [SCENE UNDERSTANDING] aims to evaluate the ability of models to understand a scene comprised of multiple views from multiple surveillance images. [VISUAL GROUNDING] aims to evaluate the ability of models to ground a specific object and seek information about it within multiple images. [VISUAL RETRIEVAL] aims to evaluate the ability of models to retrieval images that contain the same building.

Additionally, MUIRBENCH includes images covering 10 various categories of multi-image relations, such as narrative images conveying stories or ideas, ordered pages of documents and slides providing collective insights, images forming a temporal sequence presenting events, and multiple views of objects or 3D scenes offering a complete vision, with the complete distribution shown in Figure 4. These relations are summarized based on prior studies and the data collected, which reflect the common focus of the research community. In terms of image presentation, the number of images in each instance ranges from two to nine, while the input positions of images can be the beginning of question, middle of question, end of question, options, and a mix of these positions. MUIRBENCH also exhibits various image types, including but not limited to slides, maps, medical images, drone/satellite images, animations, memes, graphics, and 3D views. The data diversity from these perspectives enhances the comprehensiveness of our benchmark. More details can be found in Appendix A.

**Robust Evaluation.** Existing datasets primarily assess models' capabilities in solving answerable questions but overlook their ability to recognize what they do not know (Rajpurkar et al., 2018; Miyai et al., 2024). In real-world scenarios, there is no guarantee that user queries are answerable. A reliable multimodal LLM should directly indicate when a query is unanswerable rather than providing an answer that is most likely to be correct. In light of this, we pair each answerable instance with an unanswerable counterpart, featuring minimal differences, to provide a more robust evaluation, simulating real-world scenarios. We adopt multiple strategies to manually design the unanswerable instances, with major strategies of image replacing or reordering, question modification, and option modification introduced in Figure 5. More details can be found in Appendix A.

## 3.2 DATA COLLECTION

**Answerable Data Collection.** We invest our efforts in collecting multi-image multiple-choice question answering (MCQA) data covering various tasks and multi-image relations. Diverse data attributes enable fine-grained and diagnostic evaluation, while the multiple-choice format ensures deterministic results. To achieve this goal, we consider three sources of data, including existing datasets, dataset derivations, as well as newly collected data. *Existing data* (40.8%) come from GeneCIS (Vaze et al., 2023), SeedBench (Li et al., 2023), and IconQA (Lu et al., 2021b). *Derived data* (21.7%) reformat data into MCQA format, using multiple strategies including question generation, option rewriting, and single-image QA combination, *etc.* upon instances from NLVR2 (Suhr et al., 2019), HallusionBench (Guan et al., 2023), ISVQA (Bansal et al., 2020), and MMBench (Liu et al., 2023c). *New data* (37.5%) address certain tasks (*e.g.* geographic understanding and visual retrieval) that are underrepresented in the aforementioned collection to fulfill a more comprehensive evaluation. We manually create the question and choices for these data based on images from the National Geologic Map Database[1], University-1652 (Zheng et al., 2020; 2023), PubMed papers[2], and SciDuet slides (Sun et al., 2021). Details about curation process and data sources can be found in Appendix A.

**Unanswerable Data Collection.** As shown in Figure 5, we consider three strategies for modifying an answerable instance to its unanswerable counterpart with minimal changes. We first replace or reorder some images to disrupt the question-image and image-image relations (24.2%). We also modify the question to make it incompatible with the images and options (35.3%). In addition, we replace options to create a scenario with no correct answer (40.5%). For each answerable instance, we apply one of these three strategies. More details can be found in Appendix A.

**Quality Control.** We employ two types of quality control throughout the annotation process: automatic check with predefined rules, and a manual examination of each instance to filter out any low-quality data. The automatic check verifies valid instance format, answers, metadata values, and the coreference between image placeholders and images (ensuring no redundant image), as well as the accessibility of images. The manual examination is conducted by four experts working in this field, and filters out ambiguous queries, unclear images, and confusing instances.

## 4 EXPERIMENTS

In this section, we first describe the experimental setup and the baselines (§4.1). Then we present a comprehensive evaluation of 20 recent multimodal LLMs (§4.2). We demonstrate that while humans can answer the questions with high accuracy, MUIRBENCH is challenging for existing models. Finally, we conduct various analyses on multiple experiment settings, including sensitivity to various resolution and error analysis (§4.3).

## 4.1 EXPERIMENTAL SETUP

**Multimodal LLMs:** We evaluate MUIRBENCH on 20 recent multimodal LLMs, including models designed for considering multi-image inputs and those originally designed for single-image inputs. For multi-image input multimodal LLMs, we evaluate on GPT-4o, GPT-4-Turbo (OpenAI, 2023), Gemini Pro (Team et al., 2023), Mantis (Idefics2, clip-llama3, and siglip-llama3 versions; 8B) (Jiang et al., 2024a), VILA (v1.5-13B) (Lin et al., 2023), Idefics (9B-Instruct and v2-8B) (Laurençon et al., 2024; Laurençon et al., 2024), Emu2 (Chat) (Sun et al., 2023a) and OpenFlamingo (v2-9B) (Awadalla et al., 2023). For single-image input multimodal LLMs, we evaluate on LLaVA (v1.5, NeXT, internLM, and xtuner versions, model size 7B, 13B, and 34B) (Liu et al., 2024b; 2023b; 2024a; Team, 2023a; Contributors, 2023b), Yi-VL-6B[3], MiniGPT-4-v2 (Chen et al., 2023b), and CogVLM (Wang et al., 2023). We refer the readers to Appendix C for more details.

**Evaluation setup:** We follow the standard setup as it is in VLMEvalKit (Contributors, 2023a), where the temperature is set to 0 and retry is set to 10. For the models that do not support multiple

---

[1] `https://ngmdb.usgs.gov/ngmdb/ngmdb_home.html`
[2] `https://pubmed.ncbi.nlm.nih.gov/`
[3] More details are at the official website at `https://www.01.ai/`

| | Overall (2,600) | Counting (234) | Action. (164) | Grounding. (84) | Matching. (464) | Ordering (64) | Scene. (186) |
|---|---|---|---|---|---|---|---|
| Random Choice | 23.99 | 20.98 | 23.41 | 25.00 | 24.12 | 22.81 | 25.00 |
| Human | 93.15 | 94.87 | 97.56 | 85.71 | 94.83 | 87.50 | 94.62 |
| *Multi-Image-Trained MLLMs* | | | | | | | |
| GPT-4o (OpenAI, 2023) | 68.00 | 49.15 | 44.51 | 36.90 | 86.85 | 23.44 | 71.51 |
| GPT-4-Turbo (OpenAI, 2023) | 62.31 | 42.31 | 39.63 | 53.57 | 80.39 | 35.94 | 59.14 |
| Gemini Pro (Team et al., 2023) | 49.35 | 28.63 | 35.98 | 28.57 | 66.59 | 12.50 | 59.14 |
| Mantis-8B-Idefics2 (Jiang et al., 2024a) | 44.50 | 38.46 | 33.54 | 26.19 | 53.88 | 18.75 | 56.99 |
| Mantis-8B-clip-llama3 (Jiang et al., 2024a) | 37.38 | 29.06 | 36.59 | 21.43 | 43.32 | 18.75 | 56.99 |
| Mantis-8B-siglip-llama3 (Jiang et al., 2024a) | 36.12 | 27.35 | 37.20 | 22.62 | 43.75 | 7.81 | 54.30 |
| Idefics-9B-Instruct (Laurençon et al., 2024) | 35.43 | 29.91 | 28.05 | 13.10 | 35.99 | 12.50 | 27.41 |
| Emu2-Chat (37B) (Sun et al., 2023a) | 33.62 | 31.20 | 27.44 | 26.19 | 37.28 | 15.63 | 48.39 |
| VILA1.5-13B (Lin et al., 2023) | 33.12 | 19.66 | 28.66 | 25.00 | 40.95 | 10.94 | 56.45 |
| Idefics2-8B (Laurençon et al., 2024) | 26.08 | 21.79 | 26.22 | 26.19 | 24.78 | 15.62 | 56.45 |
| OpenFlamingo-v2-9B (Awadalla et al., 2023) | 23.73 | 21.79 | 26.83 | 30.95 | 24.14 | 21.88 | 22.58 |
| *Single-Image-Trained MLLMs* | | | | | | | |
| LLaVA-NeXT-34B (Liu et al., 2024a) | 33.31 | 36.32 | 26.22 | 33.33 | 37.93 | 21.88 | 54.30 |
| LLaVA-v1.5-7B-xtuner (Contributors, 2023b) | 33.23 | 26.92 | 25.61 | 23.81 | 22.84 | 4.69 | 39.78 |
| Yi-VL-6B [6] | 28.69 | 28.21 | 27.44 | 28.57 | 25.00 | 7.81 | 38.71 |
| LLaVA-internLM2-7B (Team, 2023a) | 28.15 | 34.19 | 26.22 | 32.14 | 25.65 | 7.81 | 42.47 |
| LLaVA-v1.5-13B (Liu et al., 2023b) | 24.38 | 25.21 | 29.27 | 14.29 | 20.26 | 20.31 | 36.56 |
| LLaVA-v1.5-7B (Liu et al., 2023b) | 23.46 | 23.08 | 27.44 | 14.29 | 23.49 | 23.44 | 34.95 |
| LLaVA-v1.5-13B-xtuner (Contributors, 2023b) | 21.69 | 23.08 | 23.17 | 16.67 | 21.98 | 14.06 | 47.85 |
| CogVLM (Wang et al., 2023) | 20.85 | 14.10 | 26.22 | 16.67 | 21.34 | 12.50 | 41.40 |
| MiniGPT-4-v2 (Chen et al., 2023b) | 17.35 | 11.97 | 14.02 | 25.00 | 17.03 | 18.75 | 14.52 |

| | Difference. (340) | Cartoon. (78) | Diagram. (398) | Geographic. (100) | Attribute. (196) | Retrieval. (292) |
|---|---|---|---|---|---|---|
| Random Choice | 23.18 | 25.00 | 29.56 | 25.00 | 20.00 | 21.30 |
| Human | 92.94 | 82.05 | 98.99 | 98.00 | 87.76 | 86.30 |
| *Multi-Image-Trained MLLMs* | | | | | | |
| GPT-4o (OpenAI, 2023) | 60.29 | 51.28 | 88.69 | 56.00 | 56.12 | 80.14 |
| GPT-4-Turbo (OpenAI, 2023) | 60.59 | 52.56 | 79.15 | 57.00 | 50.51 | 64.04 |
| Gemini Pro (Team et al., 2023) | 45.29 | 47.44 | 64.82 | 48.00 | 41.33 | 43.84 |
| Mantis-8B-Idefics2 (Jiang et al., 2024a) | 28.82 | 38.46 | 67.59 | 26.00 | 48.47 | 35.62 |
| Mantis-8B-clip-llama3 (Jiang et al., 2024a) | 24.12 | 43.59 | 54.27 | 16.00 | 33.67 | 31.85 |
| Mantis-8B-siglip-llama3 (Jiang et al., 2024a) | 27.35 | 46.15 | 47.99 | 22.00 | 31.63 | 28.08 |
| Idefics-9B-Instruct (Laurençon et al., 2024) | 34.41 | 48.72 | 46.98 | 35.00 | 32.65 | 43.49 |
| Emu2-Chat (37B) (Sun et al., 2023a) | 32.65 | 43.59 | 37.69 | 34.00 | 31.63 | 23.97 |
| VILA1.5-13B (Lin et al., 2023) | 24.71 | 30.77 | 42.71 | 31.00 | 24.49 | 30.14 |
| Idefics2-8B (Laurençon et al., 2024) | 27.65 | 39.74 | 25.38 | 21.00 | 17.86 | 17.12 |
| OpenFlamingo-v2-9B (Awadalla et al., 2023) | 21.76 | 25.64 | 31.91 | 25.00 | 18.88 | 15.41 |
| *Single-Image-Trained MLLMs* | | | | | | |
| LLaVA-NeXT-34B (Liu et al., 2024a) | 22.06 | 41.03 | 38.19 | 12.00 | 38.27 | 25.00 |
| LLaVA-v1.5-7B-xtuner (Contributors, 2023b) | 33.53 | 29.49 | 44.72 | 26.00 | 38.78 | 47.60 |
| Yi-VL-6B [6] | 25.59 | 50.00 | 35.68 | 17.00 | 34.18 | 22.60 |
| LLaVA-internLM2-7B (Team, 2023a) | 19.12 | 39.74 | 35.43 | 12.00 | 23.98 | 28.42 |
| LLaVA-v1.5-13B (Liu et al., 2023b) | 20.00 | 25.64 | 31.66 | 20.00 | 22.96 | 20.89 |
| LLaVA-v1.5-7B (Liu et al., 2023b) | 20.00 | 24.36 | 25.13 | 20.00 | 22.96 | 19.86 |
| LLaVA-v1.5-13B-xtuner (Contributors, 2023b) | 12.94 | 30.77 | 20.10 | 11.00 | 18.37 | 21.58 |
| CogVLM (Wang et al., 2023) | 19.71 | 41.03 | 19.60 | 13.00 | 16.33 | 15.75 |
| MiniGPT-4-v2 (Chen et al., 2023b) | 20.00 | 21.79 | 21.61 | 13.00 | 17.35 | 14.73 |

Table 1: **Experiment results on MUIRBENCH**. The first row shows task names and number of test data. We see that most models perform similarly to random choice, and are far from humans (§4.3).

images as input, we concatenate the images to constitute one input. Following Lu et al. (2024), our prompt consists of four parts, the question, options, the hint indicating the answer format, and a prefix indicating the beginning of the answer. For images, we insert them into the text to form a coherent prompt. Following (Yue et al., 2023), We use a rule-based automatic tool to extract the exact answer. We refer the readers to Appendix §D for more details on multi-image concatenation, visual prompting, answer extraction, and the human evaluation protocol.

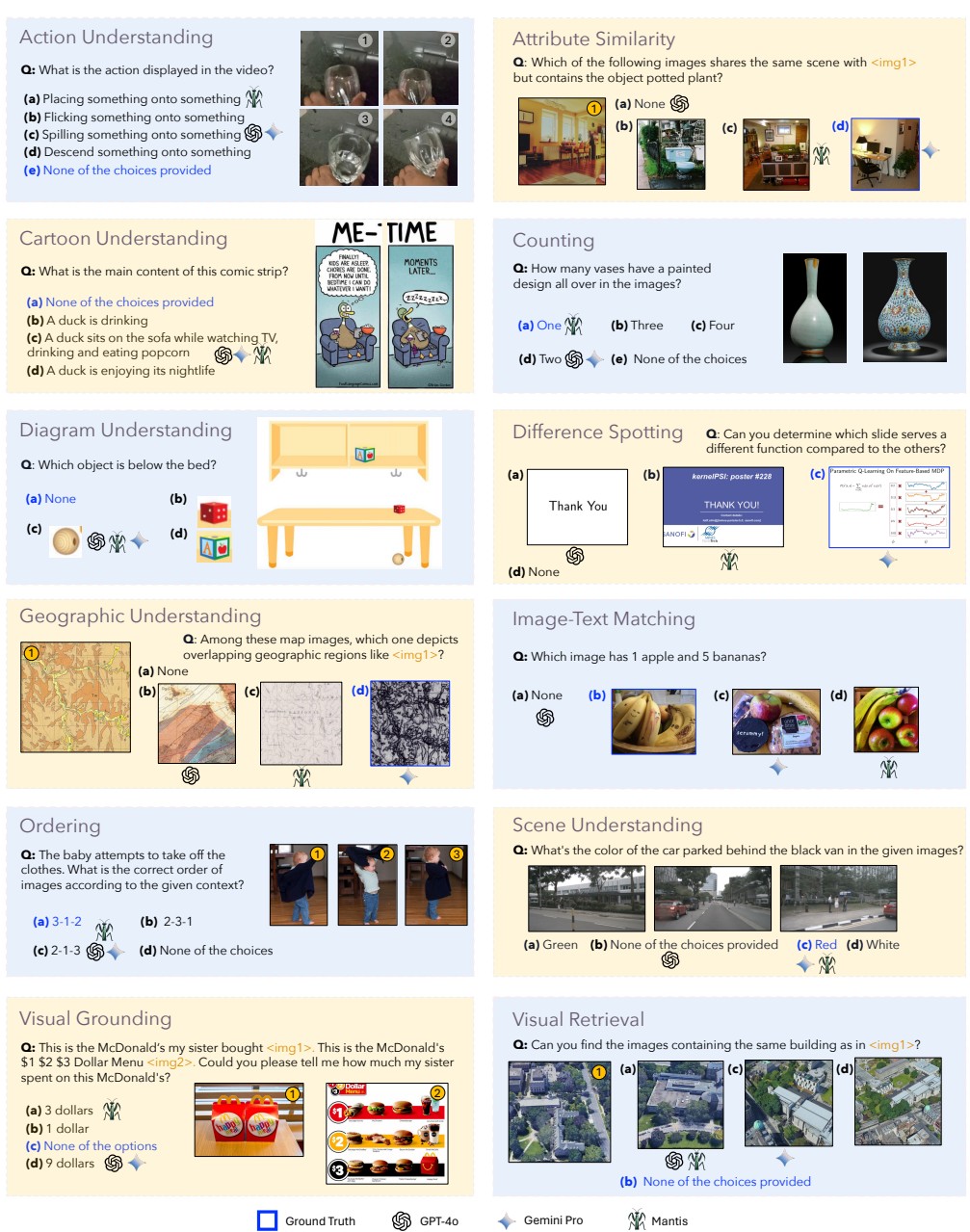

Figure 6: Qualitative results on MUIRBENCH. For each task, we show the ground-truth answer in blue, and choice of GPT-4o (OpenAI, 2023), Gemini Pro (Team et al., 2023), and Mantis-8B-Idefics2 (Jiang et al., 2024a). Notice that the example cases are slightly modified with change of word and image reduction for better illustration.

## 4.2 MAIN RESULTS

**Overall performance:** As shown in Table 1, the average accuracies of the most advanced multimodal LLMs on MUIRBENCH are no better than 68%, which are still far from enabling satisfactory utility. The mean accuracies of open-source multimodal LLMs that have considered multi-images hover between 23.73% and 44.50%, which fall behind from advanced proprietary LLMs. Notably, there is no obvious correlation between model sizes and performances, indicating the importance of training data and training processes in developing multimodal LLMs with multi-image understanding capabilities. For certain models and tasks, some results are only on par or even below random

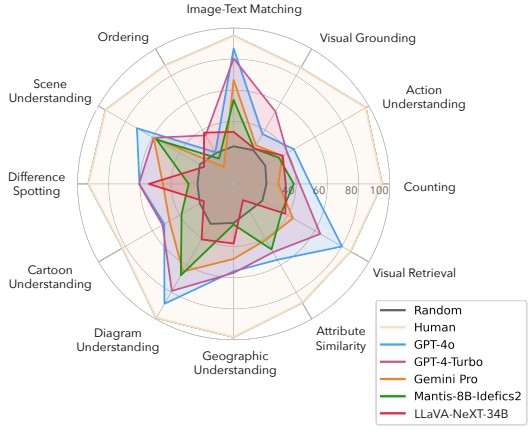

Figure 7: Model performance by tasks.

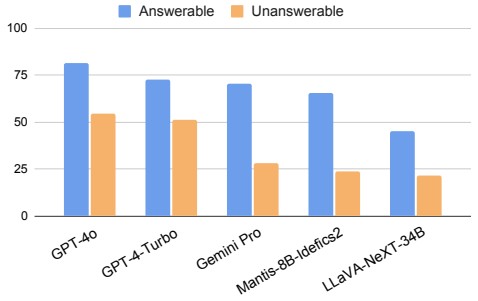

Figure 8: Model performance on answerable and unanswerable instances. An obvious performance gap can be observed between the two sets on all best-performing models.

guessing. We provide more in-depth model analyses in the following and in Appendix H. Moreover, the model performance by multi-image relation is presented in Table 3 in appendix. We also report the task-level and relation-level macro accuracy in Appendix §G.

**In which multi-image tasks do multimodal LLMs show relative strengths and weaknesses?** Figure 7 visualizes the accuracies of the best-performing models on MUIRBENCH. We observe that multimodal LLMs perform relatively better on image-text matching, visual retrieval, and diagram understanding. In contrast, multi-image ordering and visual grounding appear to be more challenging for these models, because these tasks require understanding the whole multi-image context and conducting more complicated reasoning processes across images and modalities afterwards.

**Can models designed for single-image inputs perform multi-image tasks?** In general, models accepting multi-image inputs(*e.g*., Mantis-8B), even with fewer parameters, perform better than single-image input multimodal LLMs (*e.g*., LLaVA-NeXT-34B). This observation shows that generalizing from single-image training to multi-image inference is non-trivial. Reasonably, models benefit from multi-image training data and learning processes to develop multi-image understanding capabilities.

## 4.3 ANALYSIS

**Do multimodal LLMs perform worse on the unanswerable set?** Figure 8 compares performances on answerable and unanswerable sets for some best-performing models. All the studied models have severe performance drop when changing answerable instances to unanswerable counterparts. A closer look of the error cases reveals that models often avoid abstention when facing unanswerable questions. These observations not only highlight the importance of assessing model behavior under a more realistic setting, but also show that the pairwise design improves the reliability of MUIRBENCH.

**Do image positions correlate with error rates?** We analyze the error rates of varying input positions of images and report the performance of GPT-4o, GeminiProVision, and Mantis-8B-Idefics2. As shown in Figure 9, the highest accuracy is achieved when images are positioned in options, while the highest error rate can be observed when images are in the middle of questions. This consistent trend across different models suggests that the position of images within a question correlates with the error rate. The cause of higher error rates might be that images in the middle or end of a question may interrupt the flow of context processing, increasing complexity and thus reducing model performance. It may also be attributed to the training process. These models may have seen less data with images in the middle during training.

**Do unanswerable types correlate with error rates?** We further analyze the error rates of varying unanswerable types and report the performance of the same three models in Figure 10. Results show that the error rate also correlates with the type of unanswerable instances. All the three models perform relatively better when we only change the questions to make it incompatible with original images and options. However, all models are confused when the correct option is removed and fail

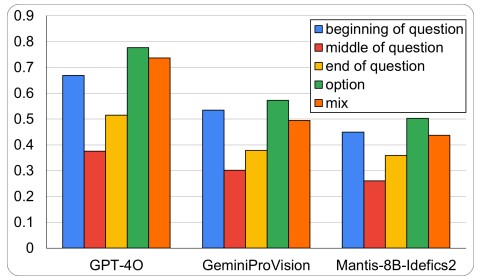 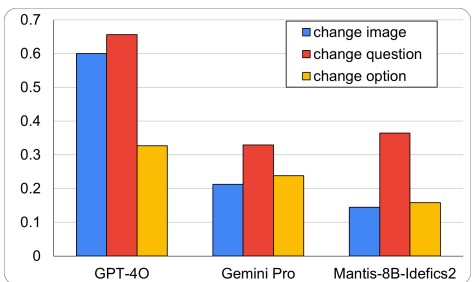

Figure 9: Model performance by image positions.    Figure 10: Performance by unanswerable types.

to choose "none of the other options" in this scenario. The performance on unanswerable instances created by reordering or replacing images is divergent. GPT-4o performs much better than the other models in these cases.

**Error analysis of GPT-4o:**  We randomly sampled 100 error instances made by GPT-4o on MUIRBENCH and meticulously examined them. The most common error category (26% of error cases) is the failure of capturing details in images. The rest 20% of errors are due to inaccurate object counting or reasoning, followed by errors in logical reasoning (18%), identification of the same object in different scenes (14%), and inferring the intents implied by image sequences (12%).

**Qualitative Results:** Figure 6 presents some qualitative results, one per task. A notable phenomenon is that multimodal LLMs may hallucinate by attempting to find an erroneous option that appears to be likely correct for an unanswerable question rather than abstaining (see examples for cartoon understanding, diagram understanding, visual grounding, and visual retrieval). This illustrates the obvious performance gap between answerable and unanswerable instances in Figure 8.

## 5    OPPORTUNITIES FOR MODEL IMPROVEMENT

Our findings highlight several opportunities for improving multimodal LLMs in multi-image scenarios. Multimodal LLMs struggle with tasks like multi-image ordering and visual grounding, which require complex reasoning across images and modalities, suggesting the need for more sophisticated training processes and model architectures that better integrate structured information of multi-image input (Wang et al., 2022). Additionally, models show weaknesses in understanding specific relations, such as temporal relation, which could be addressed by training on more temporally annotated data. Our results also reveal that models benefit from multi-image training. Thus, expanding multi-image datasets and training on diverse image types, tasks, and relations could improve generalization (Li et al., 2024a;b). Similarly, the model's performance drop on certain image positions suggests that training data should include a broader range of image positions. Furthermore, multimodal LLMs often fail to identify unanswerable questions, even though such questions are inevitable and common in the real world, pointing to the need for better training in recognizing insufficient context (Wang et al., 2024a). Lastly, the challenge of inputting multiple images often requires compression or concatenation, which can lead to information loss or long-context issues. This highlights the need for new architectures that can process multiple images more effectively, preserving context and minimizing coreference challenges.

## 6    CONCLUSION

In this work, we introduced MUIRBENCH, a comprehensive benchmark designed to provide a robust evaluation on the multi-image understanding capabilities of multimodal LLMs. Experimental results of 20 multimodal LLMs, including the prominent models like GPT-4 and Gemini Pro, revealed substantial limitations in their ability to handle multi-image scenarios. These models showed significant performance deficits compared to human accuracy and struggled more with unanswerable questions in MUIRBENCH. Our findings underscore the need for multimodal LLMs to transcend single-image limitations and achieve more holistic visual comprehension. MUIRBENCH provides a rigorous framework for such assessments, encouraging the community to develop models that can effectively synthesize and reason across multiple visual sources.

## ACKNOWLEDGMENT

Fei Wang was supported by an Amazon ML PhD Fellowship. James Y. Huang was supported by a gift fund from the USC Center on Secure & Trusted ML. Qin Liu, Wenjie Mo and Muhao Chen were supported by the DARPA FoundSci Grant HR00112490370, the NSF of the United States Grant ITE 2333736. Mingyu Derek Ma was supported by the DARPA FoundSci Grant HR00112490370. Wenxuan Zhou was supported by the NSF of the United States Grant IIS 2105329. Tianyi Yan was supported by a CURVE Fellowship. Xingyu Fu and Dan Roth were funded by ONR Contract N00014-23-1-2364. Special thanks to BLINK (Fu et al., 2024) authors, especially Wei-Chiu Ma, for providing the figure templates used in this paper.

## ETHICS STATEMENT

Our work proposes MUIRBENCH, providing a robust evaluation on multi-image tasks using multimodal LLMs. While it includes a comprehensive list of 12 tasks, all of them are in English and could induce bias on multilingual research settings. Also, if misused, the multimodal LLMs may be used to generate harmful vision and text artifacts. Nevertheless, this is not directly related to our research, and the data we curate do not contain personally identifiable information or offensive content. However, more researchers should be encouraged to get involved in research on the safety issues in a multimodal context.

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

# Appendices

## Table of Contents

## A   MUIRBENCH DETAILS

### A.1   DATASET STATISTICS

Figure 11 presents the overall statistics of MUIRBENCH. Figure 12 shows the data distribution by the type of images. MUIRBENCH covers a wide range of image types, ranging from common types like photography to specific areas such as medical images, slides, and drone and satellite imagery. Figure 13 demonstrates the data distribution by the number of images. MUIRBENCH contains instances ranging from two images to nine images. Figure 14 presents the data distribution by the position of images, including the beginning/middle/end of a question, options, and a mix of these positions.

### A.2   DATASET CURATION DETAILS

**Answerable Data Collection.** We invest our efforts in collecting multi-image multiple-choice question answering (MCQA) data covering various tasks and multi-image relations. Diverse data attributes enable fine-grained and diagnostic evaluation, while the multiple-choice format ensures

| | |
|---|---|
| Total Instances | 2600 |
| Total Images | 11264 |
| Total Tasks | 12 |
| Total Image Relations | 10 |
| Answerable Instances | 1300 |
| - existing data | 531 (40.8%) |
| - derived data | 282 (21.7%) |
| - new data | 487 (37.5%) |
| Unanswerable Instances | 1300 |
| - change image | 315 (24.2%) |
| - change question | 459 (35.3%) |
| - change option | 526 (40.5%) |
| Average image number | 4.3 |
| Average question length | 21.6 |
| Average option length | 3.7 |
| Average option number | 4.4 |

Figure 11: Statistics of MUIRBENCH.

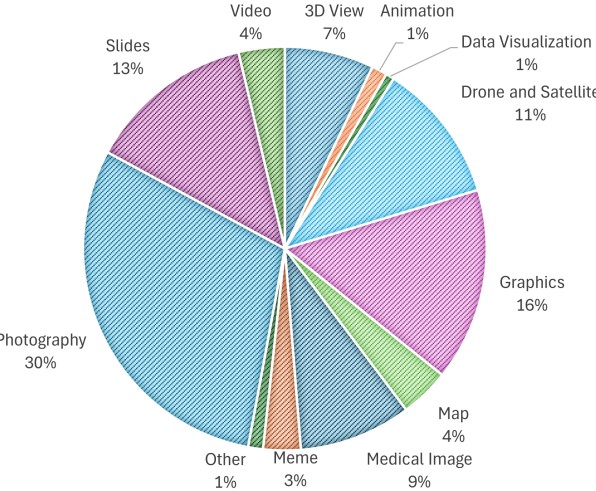

Figure 12: Data distribution by type of images.

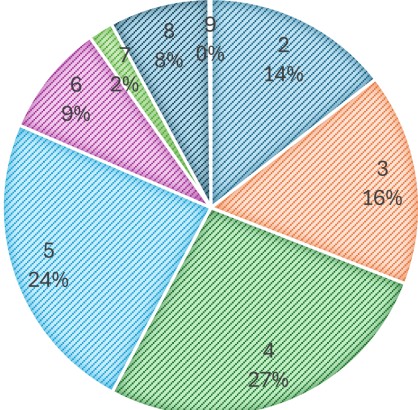

Figure 13: Data distribution by number of images.

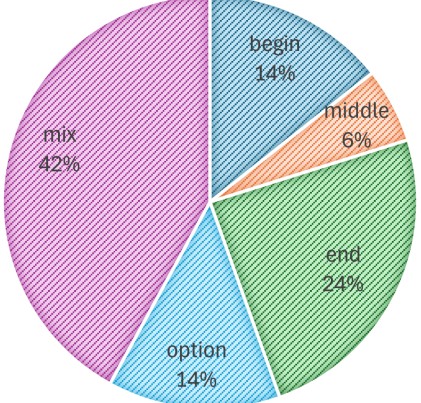

Figure 14: Data distribution by position of images.

deterministic results. To achieve this goal, we consider three sources of data, including existing datasets, dataset derivations, as well as newly collected data. *Existing data* come from datasets that focus on a single aspect of multi-image reasoning, such as GeneCIS (Vaze et al., 2023); and from datasets not specifically designed for the multi-image setting but containing a portion of multi-image data, such as SeedBench (Li et al., 2023) and IconQA (Lu et al., 2021b). For a fair representation of each task, we sample up to 200 test examples from each dataset. This part contributes 40.8% of the data in the final benchmark. *Derived data* reformat binary QA, such as NLVR2 (Suhr et al., 2019) and HallusionBench (Guan et al., 2023), into MCQA by modifying questions and options; or rewriting open QA, such as ISVQA (Bansal et al., 2020), into MCQA by adding options; and reconstructing single-image MCQA, such as MMBench (Liu et al., 2023c), into multi-image MCQA by replacing text options with corresponding images. Similar to those from the existing datasets, we sample up to 200 test examples from each dataset. This part contributes 21.7% of the data in the final benchmark.

*New data* address certain tasks (*e.g.* geographic understanding), image relations (*e.g.* multiview), and types (*e.g.* medical images) remaining absent or underrepresented in the aforementioned collection to fulfil a more comprehensive evaluation. We present four new datasets: HistoricalMap, UnivBuilding, PubMedMQA, and SciSlides. HistoricalMap requires identifying map patches covering the same regions collected from the National Geologic Map Database.[4] UnivBuilding requires identifying

---

[4] https://ngmdb.usgs.gov/ngmdb/ngmdb_home.html

different views of the same building, or buildings from the same universities. The image data are from University-1652 (Zheng et al., 2020; 2023). PubMedMQA contains questions regarding the subfigures from medical papers on PubMed.[5] SciSlides consists of questions regarding the slides for paper presentation collected from SciDuet (Sun et al., 2021). This part contributes 37.5% of the data in the final benchmark.

**Unanswerable Data Collection.** As shown in Figure 5, we consider three strategies for modifying an answerable instance to its unanswerable counterpart with minimal changes. We first replace or reorder some images to disrupt the question-image and image-image relations. We also modify the question to make it incompatible with the images and options. In addition, we replace options to create a scenario with no correct answer. For each answerable instance, we apply one of these three strategies. Among all the instances, 24.2% of the unanswerable instances are created by replacing or reordering the images in their answerable counterparts, 35.3% by modifying the questions, and 40.5% by changing the options. This step doubles the size of data, leading to a balanced distribution of answerable and unanswerable instances.

**Metadata Annotation.** Fine-grained metadata enable a diagnostic analysis of multimodal LLMs' weaknesses across various aspects. We annotate image relations, tasks, image types, number of images, and image positions for all instances. Among all of these attributes, image relations are a crucial factor that influences the model's capability for multi-image reasoning, yet they are rarely annotated in existing data. Therefore, we manually annotate them. Tasks and image types are partially annotated in existing data. We match the existing categories with our taxonomy and manually fill in any missing ones. Number of images and image positions are automatically detectable, so we conduct automatic annotation. The annotation interface is shown in Figure 16.

**Quality Control.** We employ two types of quality control throughout the annotation process: automatic check with predefined rules, and a manual examination of each instance to filter out any low-quality data. The automatic check verifies valid instance format, answers, metadata values, and the coreference between image placeholders and images (ensuring no redundant image), as well as the accessibility of images. The manual examination at last filters out ambiguous queries, unclear images, and instances with other errors, resulting in the retention of 86.3% of instances.

## A.3 MULTI-IMAGE RELATIONS

MUIRBENCH consists of 10 multi-image relations:

- *Temporal Relation*: Images are related by time, showing progression or change over a period. Examples include time-lapse photography or sequential frames from a video.
- *Ordered Pages*: Images are part of a sequence, such as pages in a book or slides in a presentation, where the order conveys meaning.
- *Complementary Relation*: Images that, when viewed together, provide additional information or context that enhances the understanding of the subject. They complement each other by filling in gaps or providing different perspectives.
- *Cropped/Zoomed Images*: One image is a zoomed-in or cropped version of another, focusing on a specific part of the original image to highlight details.
- *Narrative*: A series of images that together tell a story or convey a sequence of events, much like a comic strip or a storyboard.
- *Scene-Multiview*: Multiple images of the same scene taken from different angles or perspectives, providing a more comprehensive view of the scene.
- *Object-Multiview*: Images of the same object captured from various angles or perspectives, useful for understanding the object's three-dimensional shape.
- *Overall Similarity*: Images that are generally similar in content, style, or subject matter, but not necessarily identical. They might share common themes or visual elements.
- *Partial Similarity*: Images that share some, but not all, elements. They might have overlapping features or subjects but also contain distinct differences.

---

[5]https://pubmed.ncbi.nlm.nih.gov/

- *Independent Images*: Images that do not have a clear relation to each other. They are not connected by time, sequence, context, or content.

### A.4 HUMAN EVALUATION PROTOCOL

Two experts in domain conduct the human evaluation. Each answerable instance and its unanswerable counterparts are randomly assigned to different experts ensuring a fair evaluation. The interface for human evaluation is shown in Figure 15.

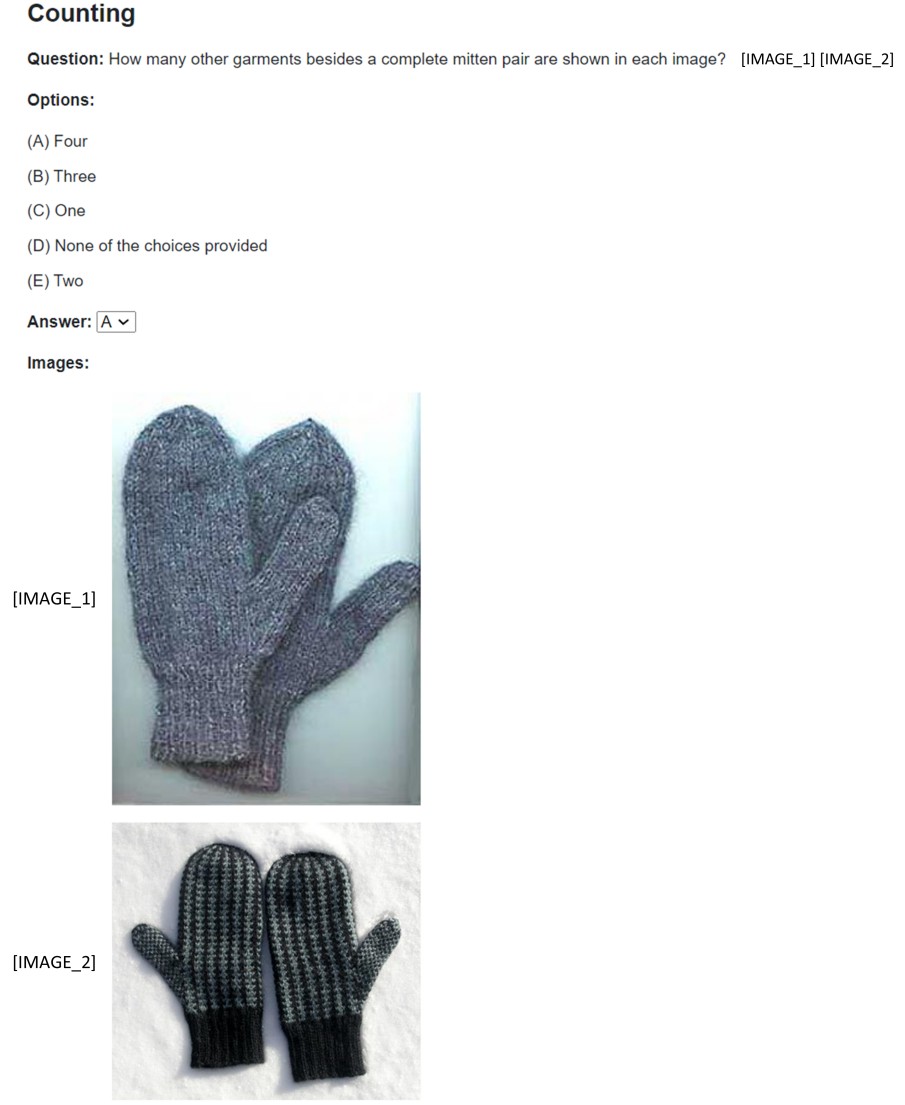

Figure 15: Human evaluation interface.

## B COMPARISON WITH RELATED BENCHMARKS

In Table 2, we compare MUIRBENCH with a series of related benchmarks.

| Benchmark | #Instances | #Image | #Relation | #Task | #Domain | Precise Metric | Unanswerable Cases |
|---|---|---|---|---|---|---|---|
| NLVR2 | 6967 | 2 | 1 | 1 | N/A | Yes | No |
| Blink | 1902 | 1-4 | N/A | 9 | N/A | Yes | No |
| MileBench-Realistic | 5197 | 2-109 | 2 | 10 | N/A | Partial | No |
| Mementos | 699 | 2-25 | 1 | 1 | 3 | No | No |
| Mantis-Eval | 217 | 2-5 | N/A | N/A | N/A | Yes | No |
| MuirBench | 2600 | 2-9 | 10 | 12 | 12 | Yes | Yes |

Table 2: Comparison with related benchmarks.

## C    BASELINE MODELS

We evaluate MUIRBENCH on 20 recent multimodal LLMs, including models designed for considering multi-image inputs and those originally designed for single-image inputs. For most model families, we use the latest and best-performing available checkpoint to date. The list of baseline models are as follows:

(i-ii) GPT-4 (OpenAI, 2023) is known to be one of the best multimodal models to date. We test with two most up-to-date checkpoints: gpt-4-turbo and gpt-4o. Notice that the GPT-4 performance would change if this specific checkpoint gets updated. (iii) Gemini Pro (Team et al., 2023) is one of the most powerful multimodal models, and we use the Gemini 1.0 Pro Vision version of it. (iv-vi) Mantis (Idefics2, clip-llama3, and siglip-llama3 versions; 8B) (Jiang et al., 2024a) is a recent strong model specifically finetuned for multi-image related tasks. (vii) VILA (v1.5-13B) (Lin et al., 2023), (viii-ix) Idefics (9B-Instruct and v2-8B) (Laurençon et al., 2024; Laurençon et al., 2024), (x) Emu2 (Chat) (Sun et al., 2023a) and (xi) OpenFlamingo (v2-9B) (Awadalla et al., 2023) are four recent multimodal models that can take multiple images as input. (xii-xvii) LLaVA (v1.5, NeXT, internLM, and xtuner versions, model size 7B, 13B, and 34B) (Liu et al., 2024b; 2023b; 2024a; Team, 2023a; Contributors, 2023b) are included as well. While they're designed for single-image input, we concatenate all the images in order. (xviii) Yi-VL-6B[6] has shown great performance recently. (xix) MiniGPT-4-v2 (Chen et al., 2023a) adapts EVA (Fang et al., 2023) as visual backbone, LLaMA2-chat (7B) (Touvron et al., 2023) as language model backbone, and designs a linear projection layer for visual understanding abilities. (xx) CogVLM (Wang et al., 2023) adds a trainable visual expert module in the attention and FFN layers to bridge different modalities better. It uses EVA-CLIP (Sun et al., 2023b) as vision encoder and Vicuna (Zheng et al., 2024) as language backbone.

## D    EXPERIMENTAL SETTING DETAILS

### D.1    MODEL PROMPTS

Following Lu et al. (2024),[7] our prompt consists of four parts, the question, options, the hint indicating the answer format, and a prefix indicating the beginning of the answer. For images, we insert them into the text to form a coherent prompt. The complete prompt is as follows:

---
**Model Prompts**

Question: {QUESTION}
Choices:
(A) {OPTION_A}
(B) {OPTION_B}
(C) {OPTION_C}
(D) {OPTION_D}
Hint: Please provide the correct option letter, such as A, B, C, D, directly.
Answer:

---

[6]More details are at the official website at `https://www.01.ai/`
[7]`https://github.com/lupantech/MathVista/blob/9ed0e8b52c0911e31faa75308082af5dcf8e63b2/evaluation/build_query.py#L152`

| | Partial Sim. | Temporal | Complementary | Scene-Multiview | Overall Sim. |
|---|---|---|---|---|---|
| gpt4o | 59.14 | 38.43 | 63.51 | 71.51 | 52.54 |
| gpt4turbo | 49.43 | 37.50 | 69.37 | 59.14 | 54.71 |
| GeminiProVision | 37.14 | 29.63 | 50.00 | 59.14 | 44.93 |
| Mantis-8B-Idefics2 | 48.00 | 29.63 | 39.19 | 56.99 | 39.49 |
| Mantis-8B-siglip-llama3 | 32.86 | 30.09 | 32.43 | 54.30 | 30.07 |
| Mantis-8B-clip-llama3 | 31.14 | 31.94 | 29.28 | 56.99 | 27.17 |
| Idefics2 | 16.57 | 23.61 | 24.77 | 56.45 | 24.64 |
| Emu2-Chat | 30.29 | 23.61 | 30.63 | 48.39 | 33.33 |
| VILA-1.5-13b | 27.43 | 23.61 | 29.28 | 56.45 | 28.26 |
| idefics1 | 31.43 | 23.61 | 30.63 | 27.42 | 31.88 |
| OpenFlamingo-9B-vitl-mpt7b | 22.86 | 25.00 | 24.77 | 23.12 | 23.55 |
| llava-v1.6-34b | 36.57 | 25.00 | 21.62 | 54.30 | 34.06 |
| llava-v1.5-7b-xtuner | 30.86 | 19.91 | 27.03 | 39.78 | 35.14 |
| Yi_VL_6B | 30.86 | 22.22 | 27.48 | 38.71 | 31.52 |
| llava-internlm2-7b | 32.29 | 21.76 | 22.97 | 42.47 | 22.10 |
| llava_v1.5_7b | 26.00 | 25.93 | 16.22 | 34.95 | 19.20 |
| llava_v1.5_13b | 25.14 | 27.31 | 17.12 | 36.56 | 19.57 |
| llava-v1.5-13b-xtuner | 24.86 | 20.83 | 13.51 | 47.85 | 15.94 |
| cogvlm-chat | 18.86 | 23.15 | 18.47 | 41.40 | 19.20 |

| | Narrative | Cropped/Zoomed | Independent | Ordered_Pages | Object-Multiview |
|---|---|---|---|---|---|
| gpt4o | 51.28 | 88.69 | 82.05 | 68.97 | 80.14 |
| gpt4turbo | 52.56 | 79.15 | 77.35 | 65.23 | 64.04 |
| GeminiProVision | 47.44 | 64.82 | 58.12 | 53.16 | 43.84 |
| Mantis-8B-Idefics2 | 38.46 | 67.59 | 46.58 | 31.90 | 35.62 |
| Mantis-8B-siglip-llama3 | 46.15 | 47.99 | 37.18 | 30.75 | 28.08 |
| Mantis-8B-clip-llama3 | 43.59 | 54.27 | 33.76 | 36.21 | 31.85 |
| Idefics2 | 39.74 | 25.38 | 18.38 | 33.33 | 17.12 |
| Emu2-Chat | 43.59 | 39.95 | 27.78 | 43.10 | 27.40 |
| VILA-1.5-13b | 30.77 | 42.71 | 37.18 | 27.87 | 30.14 |
| idefics1 | 48.72 | 47.24 | 38.03 | 32.18 | 43.49 |
| OpenFlamingo-9B-vitl-mpt7b | 26.92 | 31.91 | 21.79 | 22.99 | 15.75 |
| llava-v1.6-34b | 42.31 | 38.19 | 38.46 | 27.30 | 25.00 |
| llava-v1.5-7b-xtuner | 29.49 | 44.72 | 25.64 | 23.56 | 47.60 |
| Yi_VL_6B | 50.00 | 35.68 | 24.36 | 18.97 | 22.60 |
| llava-internlm2-7b | 39.74 | 35.43 | 24.79 | 19.54 | 28.42 |
| llava_v1.5_7b | 24.36 | 25.13 | 20.51 | 24.14 | 19.86 |
| llava_v1.5_13b | 25.64 | 31.66 | 20.09 | 22.13 | 19.52 |
| llava-v1.5-13b-xtuner | 32.05 | 20.10 | 22.22 | 14.66 | 21.58 |
| cogvlm-chat | 41.03 | 19.60 | 21.37 | 13.51 | 15.07 |

Table 3: Performance by image relation.

## D.2 EVALUATION TOOL

Following Yue et al. (2023), We use a rule-based automatic tool[8] to extract the exact answer. First, the tool detects if a valid option index appears in the model output. If no direct answer is found, the tool matches the output to the content of each option. If there is still no match, it will randomly select

---

[8] https://github.com/MMMU-Benchmark/MMMU/blob/f3e473e1e7af2c65a56ab66d7b3cf09c5dbaf0b9/eval/utils/eval_utils.py#L10

an option as the answer. When more than one valid answer is detected, the tool will use the first one that appears as the final answer.

# E    Performance by Image Relation

In Table 3, we report model performances by image relation.

# F    Performance of Recent Models

In Table 4, we further report the results of recent multimodal LLMs released after MUIRBENCH, including LLaVA-OneVision (Li et al., 2024a), InternVL2 (Chen et al., 2024b), MM1.5 (Zhang et al., 2024a), MAmmoTH-VL (Guo et al., 2024), and Gemini 1.5 Pro. Some of these works have already used MUIRBENCH as one of the primary benchmarks. The results highlight how MUIRBENCH contributes to advancing the development of multimodal LLMs.

| Model | Overall Score |
|---|---|
| LLaVA-OneVision-0.5B | 25.5 |
| LLaVA-OneVision-7B | 41.8 |
| LLaVA-OneVision-72B | 54.8 |
| InternVL2-4B | 26.8 |
| InternVL2-8B | 36.8 |
| MM1.5-1B | 34.7 |
| MM1.5-1B (MoE) | 40.9 |
| MM1.5-3B | 44.3 |
| MAmmoTH-VL-8B | 55.1 |
| Gemini 1.5 Pro (001) | 64.9 |
| Gemini 1.5 Pro (002) | 62.0 |

Table 4: Results of recent multimodal LLMs. Partial results are reported by corresponding papers.

# G    Task-Level and Relation-Level Average Performance

To obtain rebalanced results, one can also report the task-balanced or relation-balanced model performance (i.e., the macro average). We provide the reweighted performance in Table 5. While the numbers differ than the overall scores (i.e., the micro average), the trends remain roughly the same.

# H    Additional Analysis

**Can multi-image MLLMs understand single-image input?** We evaluated GPT-4o using both multiple images and a concatenated image as input for each instance. As shown in Table 6, GPT-4o performs worse when using a concatenated image as input. This result is intuitive, as GPT-4o is inherently designed to process multiple images separately. Concatenating images can lead to information loss and introduce coreference challenges between subimages and their textual mentions in the prompt.

**Can single-image MLLMs understand multi-image input?** We further evaluated LLaVA-NeXT-34B under the same setting. LLaVA-NeXT-34B exhibits a similar trend. Although primarily trained on single-image data, it has been trained using the AnyRes technique—where one large image is split into several smaller ones—allowing it to transfer this capability to handle multiple images effectively. These results suggest that explicitly processing multiple images as input is beneficial, as it reduces information loss and minimizes challenges in referencing images.

**Can the benchmark solved by dense captioning?** Following BLINK (Fu et al., 2024), we employ GPT-4o to caption each image, subsequently replacing the images with their captions as inputs. The results, presented in Table 7, show a similar trend to our prior work MMMU, which is also a benchmark for image understanding. This study specifically illustrates the extent to which details

|  | Task-Macro Avg | Relation-Macro Avg |
|---|---|---|
| GPT-4o | 58.74 | 65.63 |
| GPT-4-Turbo | 56.24 | 60.85 |
| Gemini Pro | 43.51 | 48.82 |
| Mantis-8B-Idefics2 | 39.40 | 43.35 |
| Mantis-8B-clip-llama3 | 34.14 | 36.99 |
| Mantis-8B-siglip-llama3 | 33.02 | 37.62 |
| Idefics-9B-Instruct | 32.35 | 28.00 |
| Emu2-Chat (37B) | 32.47 | 34.81 |
| VILA1.5-13B | 30.46 | 33.37 |
| Idefics2-8B | 26.65 | 35.46 |
| OpenFlamingo-v2-9B | 23.90 | 23.87 |
| LLaVA-NeXT-34B | 32.21 | 34.28 |
| LLaVA-v1.5-7B-xtuner | 30.31 | 32.37 |
| Yi-VL-6B | 28.40 | 30.24 |
| LLaVA-internLM2-7B | 27.26 | 28.95 |
| LLaVA-v1.5-13B | 23.92 | 23.63 |
| LLaVA-v1.5-7B | 23.25 | 24.47 |
| LLaVA-v1.5-13B-xtuner | 21.80 | 23.36 |
| CogVLM | 21.47 | 23.17 |

Table 5: Macro-average performance by task and relation.

| Model | Input Format | Overall Score |
|---|---|---|
| GPT-4o | multi-image | 68.00 |
|  | single-image | 60.69 |
| LLaVA-NeXT-34B | multi-image | 36.80 |
|  | single-image | 33.31 |

Table 6: Effect of input format.

beyond the provided captions are necessary to answer the questions. It does not, however, reflect the quality of the benchmark.

| Benchmark | Type | Input Modality | Score |
|---|---|---|---|
| MMBench | Multimodal Perception and Reasoning | text-only input (image caption) | 80.8 |
|  |  | multimodal input | 75.1 |
| BLINK | Visual Perception | text-only input (image caption) | 36.0 |
|  |  | multimodal input | 51.1 |
| MMMU | Multi-discipline Multimodal Understanding | text-only input (image caption) | 47.2 |
|  |  | multimodal input | 56.8 |
| MuirBench | Robust Multi-image Understanding | text-only input (image caption) | 63.4 |
|  |  | multimodal input | 68.0 |

Table 7: Effect of input modality.

# I    LIMITATION AND FUTURE WORK

There are several limitations to this work. We focus our scope on 2D images, and future research can further extend the idea of work to 3D problems, and include more multi-image tasks and relation categories. We focus on multiple-choice questions answering following widely used previous benchmark (Fu et al., 2024; Yue et al., 2023), as this format ensures structured evaluation and clear criteria for correctness. Nevertheless, other question formats, such as open-ended questions, are also valuable to explore. We hope our work can guide future efforts in providing robust and faithful evaluation in multimodal benchmarks. In addition, our strategies of creating unanswerable instances,

as in Figure 5, do not cover all strategies that can be used to create such instances. Also, we focus our evaluations on multimodal LLMs. Future work could include more vision-language foundation models such as Unified-IO 2 (Lu et al., 2023a) and Chameleon (Team, 2024).

## J  LICENSE

We release our data under CC-BY 4.0 license. For specific instances we follow their original licenses. The datasets we used and their licenses are as follows:

- *GeneCIS* is released under the CC-BY-NC 4.0 license.[9]

- *SEED-Bench* is released under the CC-BY-NC 4.0 license.[10]

- *IconQA* is released under the CC BY-NC-SA license.[11]

- *NLVR2* is released under the CC-BY-4.0 license.[12]

- *HallusionBench* is released under the BSD 3-Clause license.[13]

- *ISVQA* annotation is released under the CC BY-NC-SA 2.0 license.[14]  We only use the images from nuScenes, which is released under the CC BY-NC-SA 4.0 license.[15]

- *MMBench* is released under the Apache-2.0 license.[16]

- *National Geologic Map Database* is free in the public domain.[17]

- *University-1652* is released under the MIT license.[18]

- *PubMed* is a free and public database, with open access articles under a Creative Commons or similar license.[19]

- *SciDuet* is released under the Apache 2.0 license with paper slides from ACL, ICML, and NeurIPS.[20]

## K  ACCESSIBILITY OF MUIRBENCH

### K.1  DATASET DOCUMENTATION AND FORMAT

The full documentation of MUIRBENCH is on the project page at `https://huggingface.co/datasets/MUIRBENCH/MUIRBENCH`. For each data entry in MUIRBENCH, it includes metadata of index (idx), task, question, options, answer, image relation, image type, images, and counterpart instance idx.

---

[9]`https://github.com/facebookresearch/genecis/tree/main?tab=readme-ov-file#license`

[10]`https://huggingface.co/datasets/AILab-CVC/SEED-Bench`

[11]`https://iconqa.github.io/`

[12]`https://github.com/lil-lab/nlvr/tree/master?tab=readme-ov-file#licensing`

[13]`https://github.com/tianyi-lab/HallusionBench?tab=readme-ov-file#license`

[14]`https://github.com/ankanbansal/ISVQA-Dataset/tree/master?tab=License-1-ov-file`

[15]`https://www.nuscenes.org/terms-of-use`

[16]`https://github.com/open-compass/MMBench?tab=Apache-2.0-1-ov-file`

[17]`https://www.usgs.gov/faqs/what-are-terms-uselicensing-map-services-and-data-national-map`

[18]`https://github.com/layumi/University1652-Baseline?tab=MIT-1-ov-file#readme`

[19]`https://www.ncbi.nlm.nih.gov/pmc/about/copyright/`

[20]`https://github.com/IBM/document2slides?tab=Apache-2.0-1-ov-file`

### K.2 LINKS AND MAINTENANCE PLAN

MUIRBENCH is hosted on Huggingface/Datasets,[21] where license and metadata[22] are also available. We maintain our benchmark on this page and will continually update it. The evaluation code and outputs will be provided to facilitate easy reproduction and analyses of the results in the paper.

### K.3 AUTHOR STATEMENT

We confirm that we bear all responsibility in case of violation of rights during the collection of data on MUIRBENCH, ensuring accountability and commitment to maintaining ethical standards. We will take appropriate action when needed.

### K.4 INTENDED USES

The dataset is for academic purposes only and not for commercial usage.

---

[21]https://huggingface.co/datasets/MUIRBENCH/MUIRBENCH
[22]https://huggingface.co/api/datasets/MUIRBENCH/MUIRBENCH/croissant

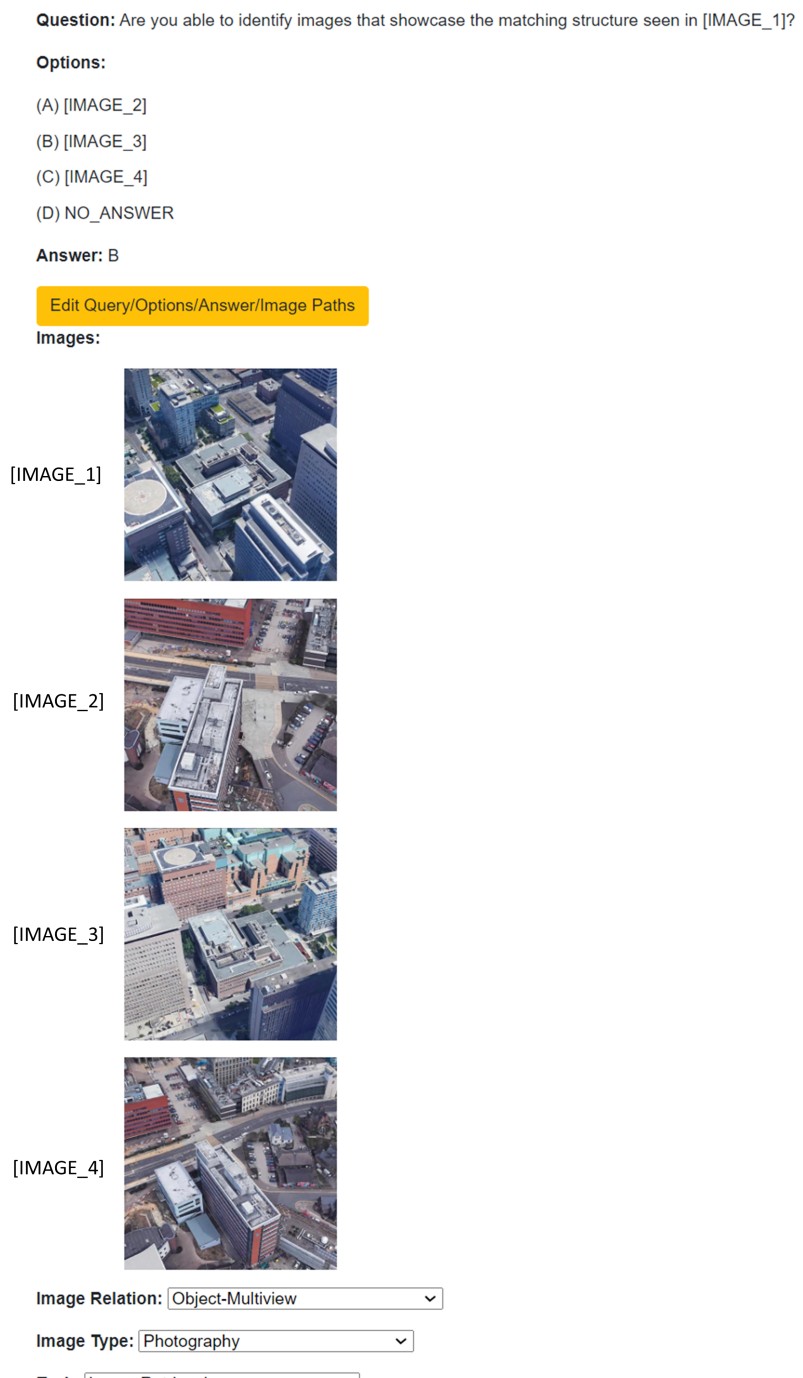

Figure 16: Annotation interface.

