# OpenReview forum: "MuirBench: A Comprehensive Benchmark for Robust Multi-image Understanding"
_ICLR.cc/2025/Conference — ICLR 2025 Poster_

### Official Review · Reviewer_1gpk · 2024-10-20

**Soundness:** 2
**Presentation:** 2
**Contribution:** 1
**Rating:** 3
**Confidence:** 5

**Summary:**

The paper introduces MuirBench mainly focusing on robust multi-image understanding capabilities of VLMs. The benchmark covers 12 multi-image understanding tasks and introduces unanswerable data. 20 recent VLMs are tested on the benchmark and several related analyses are provided.

**Strengths:**

1. The MuirBench covers 12 multi-image understanding tasks, involving diverse scenarios. It is a comprehensive benchmark containing 11,264 images and 2,600 multiple-choice questions.
2. The paper conducts detailed testing on 20 recent VLMs, revealing the limitations of these models in multi-image understanding and the performance gap compared to single-image input.
3. The analyses of unanswerable-type questions, image positioning, and error analysis of GPT-4o are impressive.

**Weaknesses:**

1. The paper only contributes data without offering any insightful analysis of model limitations and lacks novelty and contribution. It does not give people any new knowledge, and people do not feel any insight after reading it.
2. Authors overlooked some of the newer VLMs, such as Gemini 1.5 Pro (Feb. 2024), InternVL1.5-Chat (Apr. 2024) and Claude 3.5 Sonnet (Jun. 2024)
3. Although some limitations of the models in multi-image understanding tasks are shown, there is no discussion on the architectural limitations of these models, why these VLMs underperformed, why certain models perform poorly on different tasks or how models’ architectures could be adapted for better multi-image understanding. Furthermore, the suggestions for future improvements are rather general and lack detailed explanations of specific technical approaches for new VLMs. To sum up, the analysis part in this paper is superficial, more insight is required.
4. The difficulty of question among different tasks is imbalance.
5. It seems that unanswerable questions have limited practical applications in real-world scenarios.
6. Some of the questions in the benchmark do not need to be presented in multiple images, such as question “Counting” and “Image-Text Matching” in Figure 6. As a questioner, I would consider inputting images one by one to ensure accuracy. A benchmark needs to focus more on practical application value rather than just creating a question that is difficult for VLMs.

**Questions:**

1. What specific shortcomings of the VLMs’ capabilities do your benchmark actually reveal?
2. Authors are encouraged to provide results on some of the latest models, such as InternLM-XComposer2.5-7B (2024/07/03), the InternVL2 series (2024/07/04), and the Qwen2-VL series (2024/08/30). While the results of these models are not mandatory under the guidelines, considering the super-fast advancements in VLMs this year. Could you please include results from some of the aforementioned models to highlight the performance of the latest generation of VLMs in multi-image understanding?
3. The pairwise design of answerable and unanswerable questions seems to create an artificial scenario. Unanswerable questions are more ambiguous and complex. How does your dataset account for the nuance of real-world multi-image understanding?
4. Line 311 mentions "concatenate the images to constitute one input." How exactly are the images concatenated? Could it be that the concatenation method affects the model's performance? What are the results of using different concatenation methods?
5. Please refer to Weaknesses for more potential questions.

---

> ### Author Response · Authors · 2024-11-24
>
> We thank the reviewer for the feedback and provide a detailed response below.
>
> > **W1. The paper only contributes data without offering any insightful analysis and lacks novelty and contribution.**
>
> We want to clarify that, for a benchmark paper, **the evaluation data and framework already constitutes a substantial contribution**. Prior to our work, there was no comprehensive benchmark for this specific scenario. Our benchmark fills this gap, **providing a testbed** for evaluating the strengths and weaknesses of models and **enabling future analyses** from various perspectives of multimodal LLMs.
>
> Our paper is **the first to provide a comprehensive benchmark** for evaluating robust multi-image understanding capabilities in multimodal LLMs. We developed a benchmark encompassing 12 tasks and 10 image relations, **offering in-depth analyses that can guide future research**. Specifically, our work identifies gaps in current model capabilities and provides inspiration for researchers to expand multi-image datasets using our data collection methodology. **To the best of our knowledge, this is the first paper that delivers such insights and contributions.** We provide a comprehensive list of analyses and insights in the following responses.
>
> > **W3/Q1. the analysis part in this paper is superficial, more insight is required.**
>
> We respectfully disagree that the analyses are superficial. We have provided **in-depth analyses in Section 4 and Appendix D**. All of these analyses provide valuable insights. We have analyzed model performance on the unanswerable set, error cases, and qualitative results. We have further examined the influence of image position, unanswerable type, and input format. We also analyzed the role of captioning in solving multi-image problems. **Researchers will gain insight on how to advance the development of multimodal LLMs**, such as (1) task, relation, and question formats to cover in the training data, (2) the tradeoff between resolution and long visual context, and (3) enhancing robustness and preventing prediction shortcuts in the context of multi-image understanding.
>
> Below is a selected list of analyses and findings in this paper:
> * In which multi-image tasks do multimodal LLMs show relative strengths and weaknesses?
>   * We find that multi-image ordering and visual grounding appear to be more challenging for existing MLLMs, because these tasks require understanding the whole multi-image context and conducting more complicated reasoning processes across images and modalities afterwards. (lines 448-450)
> * In which multi-image relations do multimodal LLMs show relative strengths and weaknesses?
>   * We find that existing MLLMs fall short in understanding temporal relations and identifying cross-image similarity. (Table 2)
> * Can models designed for single-image inputs perform multi-image tasks?
>   * Our results show that models benefit from multi-image training data and learning processes to develop multi-image understanding capabilities. (lines 455-456)
> * Do multimodal LLMs perform worse on the unanswerable set?
>   * Our results reveals that models often fail to identify unanswerable questions, indicating that models may not comprehensively understand the multi-image context. (lines 463-465)
> * Do image positions correlate with error rates?
>   * Our results emphasize the necessity of training data covering diverse image positions to improve model generalizability. (lines 470-474)
> * Do unanswerable types correlate with error rates?
>   * This analysis demonstrate the distinct weaknesses of different MLLMs on this problem. (lines 477-480)
> * Error analysis of GPT-4o
>   * We identify the key weaknesses of GPT-4o in multi-image understanding, including capturing details, counting, and logical reasoning. (lines 485-500)
> * Can multi-image MLLMs understand single-image input?
>   * This analysis shows that concatenating images can lead to information loss and introduce coreference challenges between subimages and their textual mentions in the prompt. (lines 1065-1067)
> * Can single-image MLLMs understand multi-image input?
>   * The results suggest that explicitly processing multiple images as input is beneficial, as it reduces information loss and minimizes challenges in referencing images. (lines 1072-1074)
>
> Our work identifies important problems of robust multi-image understanding in multimodal LLMs and presents a way to quantitatively evaluate the corresponding capabilities.  In terms of  **model architecture**, our analyses suggest to **go beyond the current sequential input structure and attention mechanism** to better capture the diverse relations across multiple images.

---

> ### Author Response · Authors · 2024-11-24
>
> > **W2/Q2. Authors overlooked some of the newer VLMs, such as Gemini 1.5 Pro, InternVL1.5, ...**
>
> We report the results of multiple most recent VLMs, including LLaVA-OneVision (the subsequent version of LLava-Interleave; released in Aug. 2024), InternVL2 (the subsequent version of InternVL1.5; released in Jul. 2024), and Gemini-1.5-Pro (version 002 released in Sept. 2024).
>
> | Model | Score |
> | - | -|
> | LLaVA-OneVision (0.5B) | 25.5 |
> | LLaVA-OneVision (7B) | 41.8 |
> | LLaVA-OneVision (72B) |  54.8 |
> | InternVL2-4B | 26.8 |
> | InternVL2-8B | 36.8 |
> | Gemini 1.5 Pro (001) | 64.9 |
> | Gemini 1.5 Pro (002) | 62.0 |
>
> > **W4. The difficulty of question among different tasks is imbalance.**
>
> We want to clarify that question difficulty is **not the primary focus** of this benchmark. Our goal is to evaluate multi-image understanding capabilities comprehensively and robustly, rather than to explicitly measure or categorize the difficulty of individual queries. Moreover, there is currently **no widely accepted standard for quantifying query difficulty** in the context of multimodal benchmarks. Attempting to do so would introduce additional subjectivity and complexity, potentially detracting from the primary objectives of our work.
>
> Nevertheless, to facilitate fine-grained analysis of model performance, we have reported the performance on **each task** (Table 1) and **each multi-image relation** (Table 2) . This follows the common practice of many widely-used benchmarks, such as BLINK and MathVista.
>
> To obtain rebalanced results, one can also report the **task-balanced** or **relation-balanced** model performance (i.e., the macro average). We provide the reweighted performance below. While the numbers differ, the trends remain roughly the same.
>
> |                                                     | Avg   | Task-Macro Avg | Relation-Macro Avg |
> | --------------------------------------------------- | ----- | -------- | ------------ |
> | GPT-4o                                 | 68.00    | 58.74    | 65.63        |
> | GPT-4-Turbo                            | 62.31 | 56.24    | 60.85        |
> | Gemini Pro                  | 49.35 | 43.51    | 48.82        |
> | Mantis-8B-Idefics2          | 44.50  | 39.40    | 43.35        |
> | Mantis-8B-clip-llama3      | 37.38 | 34.14    | 36.99        |
> | Mantis-8B-siglip-llama3     | 36.12 | 33.02    | 37.62        |
> | Idefics-9B-Instruct | 35.43 | 32.35    | 28.00        |
> | Emu2-Chat (37B)          | 33.62 | 32.47    | 34.81        |
> | VILA1.5-13B                     | 33.12 | 30.46    | 33.37        |
> | Idefics2-8B                       | 26.08 | 26.65    | 35.46        |
> | OpenFlamingo-v2-9B | 23.73 | 23.90    | 23.87        |
> | LLaVA-NeXT-34B             | 33.31 | 32.21    | 34.28        |
> | LLaVA-v1.5-7B-xtuner            | 33.23 | 30.31    | 32.37        |
> | Yi-VL-6B                   | 28.69 | 28.40    | 30.24        |
> | LLaVA-internLM2-7B             | 28.15 | 27.26    | 28.95        |
> | LLaVA-v1.5-13B         | 24.38 | 23.92    | 23.63        |
> | LLaVA-v1.5-7B         | 23.46 | 23.25    | 24.47        |
> | LLaVA-v1.5-13B-xtuner            | 21.69 | 21.80    | 23.36        |
> | CogVLM                    | 20.85 | 21.47    | 23.17        |
>
> > **W5. It seems that unanswerable questions have limited practical applications in real-world scenarios.**
>
> Unanswerable questions are **practical and integral to assessing model robustness**, as outlined in the introduction of our paper and **supported by prior research** [1,2]. This aspect is previously overlooked in multi-image scenarios but is essential for understanding how models handle uncertainty and incomplete information. **Our results (Figure 8) underscore the significance** of this feature for a comprehensive evaluation of model performance. The significant performance gap between answerable and unanswerable sets indicates that models may achieve good performance by cheating.
>
> Moreover, the pairwise design (answerable-unanswerable) is intended to ensure that the model cannot achieve good performance by cheating. Models may answer questions correctly by making an educated guess without fully understanding all the images. To avoid this, we **pair each answerable instance with an unanswerable counterpart**. This **ensures robustness and reliability in evaluating model performance**.
> According to our analysis in Figure 8, the top five models achieve significantly better performance on the answerable set than on the unanswerable set. This highlights the challenges of ensuring robust understanding rather than leveraging shortcuts.
>
> [1] Rajpurkar, Pranav, Robin Jia, and Percy Liang. "Know What You Don’t Know: Unanswerable Questions for SQuAD." ACL 2018.
>
> [2] Miyai, Atsuyuki, et al. Unsolvable Problem Detection: Evaluating Trustworthiness of Vision Language Models.

---

> ### Author Response · Authors · 2024-11-25
>
> > **W6. As a questioner, I would consider inputting images one by one to ensure accuracy.**
>
> Note that this is a benchmark, while **the reviewer is indeed discussing how to improve performance** through preprocessing or inference-time techniques. **This actually highlights the value of our benchmark** in inspiring and supporting such research.
>
> In reality, **model developers cannot assume how users will present their questions**. The reviewer’s method is only one proposal for question formatting, but questions can naturally appear in diverse formats in real-life usage. Our goal is to collect a comprehensive benchmark that measures robust multi-image understanding capabilities, regardless of how users present their questions.
>
> > **Q4. How exactly are the images concatenated?**
>
> We concatenate the images following the setting of previous work [1]. The concatenation method is not the focus of our work. Instead, we seek to provide a testbed to analyze every aspect of multi-image VLMs, including the one mentioned by the reviewer. We leave specific concatenation testing analyses for future research using our benchmark.
>
> [1] Jiang, Dongfu, et al. "Mantis: Interleaved multi-image instruction tuning." arXiv preprint arXiv:2405.01483 (2024).

---

> > ### Author Response · Authors · 2024-11-26
> >
> > Dear Reviewer 1gpk,
> >
> > Thank you for your time and detailed feedback on our paper. We hope our responses have addressed your concerns, and we kindly request that you consider updating the score accordingly. If there are any remaining issues, please let us know, and we will be happy to provide further clarification.

---

> > ### Comment · Reviewer_1gpk · 2024-11-26
> > **Reply to authors**
> >
> > I sincerely appreciate your hard work and effort, some of my concerns are addressed. However, I still have several concerns.
> >
> > For W2/Q2, W4. Thank you so much!
> >
> > For W3/Q1, W1. Thank you so much for providing a comprehensive list of your analyses and findings in this paper, however, as you mentioned, many of the analyses primarily describe the observed phenomena without delving into actionable insights. After reading this paper, I find myself getting access to a comprehensive small scale dataset only get the dataset and limited concrete strategies or guidance that may help improve the performance of MLLMs in multi-image understanding.
> >
> > For W5. Part of my concerns of unanswerable questions setting has been addressed, it is indeed practical and integral to assessing model robustness. Additionally, I also agree with [Reviewer QRSh](https://openreview.net/forum?id=TrVYEZtSQH&noteId=6ZfWpqCoor), it can be easier and not a key contribution.
> >
> > For W6. The point I would like to discuss with you is whether changing the question format can significantly enhance the performance of MLLMs. If that is the case, it may diminish the value of your benchmark. I would appreciate if you could share more related experimental results to verify the value of your benchmark.
> >
> > For Q4. Thank you for your clarifying. For example, InternVL series concatenate images along the horizontal direction, Qwen2-VL series concatenate images at the token level, different concatenation methods will lead to different performance for different models. I tend to believe that this task should also fall within the scope of robust multi-image understanding. I would appreciate if you can add more related experimental results into the supplementary section.
> >
> > Thank you so much for your comprehensive response!

---

> ### Author Response · Authors · 2024-11-26
>
> We appreciate the reviewer’s active discussion and are happy to provide further clarification as follows.
>
> > **For W3/Q1, W1. many of the analyses primarily describe the observed phenomena without delving into actionable points**
>
> We want to clarify that our contribution, analyses, and data size are **comparable to previous widely-used multimodal benchmarks** with different focuses. The **key issue** is that, prior to our benchmark, researchers didn’t even have a comprehensive and rigorous metric to assess the multi-image understanding capabilities of VLMs.
>
> This paper, like any benchmark paper, focuses on identifying bottlenecks and providing a rigorous evaluation framework. The scope of analyses we provided are similar to papers of widely used benchmarks, such as BLINK (ECCV 2024), MMMU (CVPR 2024), MathVista (ICLR 2024), etc. Note that fine-grained analysis for actionable improvements is also not the primary focus of these papers.
>
> In terms of data scale, the current data size (2,600) is sufficient to advance the development of VLMs. For example, datasets like Winoground (CVPR 2022), which contains only 400 examples, have significantly influenced VLM advancements. Similarly, although focused on different aspects, other widely adopted benchmarks—including WHOOPS! (ICCV 2023), LlaVA-Bench (NeurIPS 2023), Visit-Bench (NeurIPS 2024), ConTextual (ICML 2024), VibeEval, and Visual Riddles (NeurIPS 2024)—comprise 90, 500, 576, 500, 269, and 400 examples, respectively, and have been pivotal for evaluating VLMs.
>
> We argue that proposing ideas for improving model capabilities and providing supporting experiments are **beyond the scope of any benchmark paper**, as the purpose of such work is to assess and highlight limitations, not to propose solutions. Including this would be unreasonable **within the constraints of a 10-page paper**, especially given our already extensive appendix making the paper 24 pages in total.
>
> > **For W5. unanswerable questions can be easier and not a key contribution**
>
> We want to clarify that the randomizing answer choices mentioned by reviewer QRSh **can not address the robustness concerns identified** in this paper. Randomizing answer choices is only shown to be effective in reducing position bias. (The answer choices in our MuirBench are already randomly shuffled after data creation, ensuring fairness in evaluation.) There are more critical biases in model evaluation, such as overfitting to specific patterns or failing to handle unanswerable questions effectively. Our approach deliberately includes unanswerable questions to assess the model's ability to abstain when appropriate, which is crucial for robust and reliable evaluation.
>
> Additionally, identifying the poor performance of VLMs on unanswerable questions **offers further insights** into their current bottlenecks. This finding highlights critical areas for improvement and emphasizes the need for more robust reasoning capabilities in VLMs. To address this problem, researchers could incorporate more realistic unanswerable data into the training process.
>
> > **For W6. whether inputting images one by one can significantly enhance the performance of MLLMs**
>
> We refer the reviewer to **Table 4**, also presented below. In this experiment, we evaluate an alternative questioning format: first, GPT-4o extracts key information from each image independently (i.e., one by one), followed by a query to answer the question based on the extracted information. The results show that this format **does not improve performance**. Instead, the additional steps amplify error propagation. Furthermore, processing images independently introduces efficiency concerns, as the number of API calls scales linearly with the number of images, making it impractical in real-world applications.
>
> | Input Format | Score |
> | - | - |
> | Independent Images | 63.4 |
> | Multi-image | 68.0 |
>
> > **For Q4. more related experimental results on different concatenation methods**
>
> We provide an analysis in Appendix D that investigates the effects of concatenating image tokens versus concatenating image pixels. The results, shown in **Table 3**, reveal performance variance across different methods, highlighting the value of our benchmark in facilitating such analyses.
>
> However, we argue that while additional experiments could always be conducted, they are not the primary focus of our benchmark.
>
> | Model | Input Format | Score |
> | - | - | - |
> | LLaVA-NeXT-34B | Concatenated Image | 33.31 |
> | LLaVA-NeXT-34B | Multiple Images (Concat image token) | 36.80 |
> | GPT-4o | Concatenated Image | 60.69 |
> | GPT-4o  | Multiple Images | 68.00 |

---

> > ### Author Response · Authors · 2024-11-27
> >
> > ### **Clarification of Our Analysis Scope**
> >
> > We wish to clarify that **the depth and scope of analysis in our paper are sufficient for a benchmark study.** We believe that the provided analysis is both insightful and impactful, offering valuable guidance for advancing MLLM development.
> >
> > ---
> >
> > ### **Addressing the Concern on Analysis Depth**
> >
> > Since the primary concern revolves around the depth of analysis, we draw a direct comparison with the papers proposed by the reviewer. This comparison demonstrates that **our analysis is at least on par and, in some aspects, more comprehensive than previous benchmarks, such as M3CoT (ACL 2024) and MathVerse (ECCV 2024).**
> >
> > ---
> >
> > ### **Comparison with M3CoT**
> >
> > Below, we outline our one-to-one comparisons to emphasize the scope of our analysis relative to M3CoT.
> >
> > - **M3CoT:** Multi-modal tool usage on text-modal LLMs fails.
> >   **MuirBench:** MLLMs trained mainly on single-image data perform poorly on multi-image understanding (Lines 452–456).
> >
> > - **M3CoT:** Performance cannot be boosted by text-only examples.
> >   **MuirBench:** Performance can drop when captioning images independently (Lines 1075–1079).
> >
> > - **M3CoT:** Performance may even be harmed by interleaved image and text examples.
> >   **MuirBench:** Error rate is correlated with image position (Lines 466–474).
> >
> > - **M3CoT:** Fine-tuning on M3CoT leads to better performance, and VLLMs benefit more than traditional VLMs.
> >   **MuirBench:** MLLMs trained with multi-image data perform significantly better (Lines 372–377).
> > ---
> >
> > ### **Comparison with MathVerse**
> >
> > Below, we outline our one-to-one comparisons to emphasize the scope of our analysis relative to MathVerse.
> >
> > - **MathVerse:** MLLMs rely more on diagram interpretation (DI) than seeing diagrams.
> >   **MuirBench:** MLLMs may rely on shortcuts and fail to identify unanswerable instances (Lines 460–465).
> >
> > - **MathVerse:** MLLMs are moderately effective at perceiving implicit premises (IP).
> >   **MuirBench:** MLLMs perform relatively better on image-text matching and visual retrieval (Lines 429–431).
> >
> > - **MathVerse:** MLLMs struggle to interpret explicit concepts (EC) from diagrams.
> >   **MuirBench:** Tasks requiring understanding multi-image contexts and inter-image relations are more challenging (Lines 448–450).
> >
> > - **MathVerse:** MLLMs struggle to solve problems entirely using diagrams.
> >   **MuirBench:** MLLMs struggle to fully comprehend multi-image contexts.  (Lines 372-377)
> >
> > - **MathVerse:** Closed-source MLLMs perform better.
> >   **MuirBench:** Open-source models lag behind proprietary ones (Lines 374–375, Table 1).
> >
> > - **MathVerse:** LLMs achieve competitive results compared to MLLMs.
> >   **MuirBench:** Multi-image understanding cannot be solved through dense captioning (Lines 1075–1079, Table 4).
> >
> > - **MathVerse:** GPT-4(V) outperforms humans on the text-only version.
> >   **MuirBench:** Advanced MLLMs, such as GPT-4o, still fall short of satisfactory utility (Lines 372–373, Table 1).
> >
> > - **MathVerse:** Mathematical training improves performance.
> >   **MuirBench:** Models trained with multi-image data achieves better performance (Lines 452–456, Table 1).
> >
> > ---
> >
> > ###  **Conclusion**
> > The above comparisons demonstrate that the scope and depth of our analysis are comparable to those in MathVerse and M3CoT, both of which are regarded as impactful benchmarks. Additionally, we provide novel insights tailored to multi-image understanding, which distinguishes MuirBench from prior works. We hope this clarifies the value and comprehensiveness of our analysis.

---

> > > ### Author Response · Authors · 2024-11-27
> > >
> > > Dear reviewer 1gpk,
> > >
> > > To address your concerns regarding actionable insights, we have **added the following discussion on the opportunities for model improvement in Section 5** of the updated paper.
> > >
> > > ---
> > >
> > > **Opportunities for Model Improvement**
> > >
> > > Our findings highlight several opportunities for improving multimodal LLMs in multi-image scenarios. Multimodal LLMs struggle with tasks like multi-image ordering and visual grounding, which require complex reasoning across images and modalities, suggesting the need for more sophisticated training processes and model architectures that better integrate inter-image relationships. Additionally, models show weaknesses in understanding specific relations, such as temporal relation, which could be addressed by training on more temporally annotated data. Our results also reveal that models benefit from multi-image training. Thus, expanding multi-image datasets and training on diverse image types, tasks, and relations could improve generalization. Similarly, the model's performance drop on certain image positions suggests that training data should include a broader range of image positions. Furthermore, multimodal LLMs often fail to identify unanswerable questions, which are inevitable and common in the real world, pointing to the need for better training in recognizing insufficient context. Lastly, the challenge of inputting multiple images often requires compression or concatenation, which can lead to information loss or long-context issues. This highlights the need for new architectures that can process multiple images more effectively, preserving context and minimizing coreference challenges.

---

### Official Review · Reviewer_QRSh · 2024-10-30

**Soundness:** 2
**Presentation:** 3
**Contribution:** 2
**Rating:** 5
**Confidence:** 5

**Summary:**

The paper proposes MuirBench, a benchmark that comprehensively measures LVLM's multi-image understanding capabilities, including multiple image relationships and multiple task types, and conducts a detailed evaluation and analysis on MuirBench.

**Strengths:**

1. The author proposed MuirBench, which is a comprehensive enough benchmark to measure the multi-graph understanding ability of the model
2. The author conducted a relatively detailed evaluation and discussion, which can provide some insights for the community

**Weaknesses:**

1. The first picture of Figure5 seems to be ambiguous: the correct order can be washing dishes and then opening the cabinet, but the reordered order can be interpreted as washing dishes, opening the cabinet and then closing it, depending on the frequency and method of frame extraction. When the number of pictures is small, reordering does not necessarily make the problem unanswerable. (Just like when looking for patterns, if there is not enough prior information, there are many ways to explain the pattern)

2. Unanswerable setting experiment may be ambiguous: Imagine such a scene: when a poor student faces a difficult question, he does not understand the question at all and cannot make a choice. At this time, the student may choose randomly or refuse to answer because he thinks he lacks the necessary information (the large model also has the phenomenon of complete question information, but refuses to answer due to insufficient ability [1]). If we change the question information to make the question unanswerable, and add unanswerable options. At this time, facing the unanswerable option, the poor student may directly choose the unanswerable option, resulting in "cheating". However, we cannot assume that the student understands the question. Similarly, the author points out that most models have poor multi-image understanding performance, which can be compared to the poor students mentioned above, so I have doubts about the settings and results of unanswerable.

3. There are problems with the settings of Multi-Image Input Models: I don’t understand the sentense: “the models that do not support multiple images as input” in 310. For the “SIngle-Image Input Models" defined by the author here, why can’t we concat the image tokens and then input them to the subsequent modules? Like InternVL1.5-chat [2], GLM-4V-9b. It has not been trained with multi-image data, but it can still perform multi-image QA through the above operations. Or should the author’s definition of “Multi-Image Input Models” here be models trained with multi-image data?

4. Although the author used the method of directly concat input images to test "single-image input models" here, this will result in too low image resolution, which may affect the performance of these models. The author should compare the effect of testing the "single-image input models" defined here using concat visual tokens. For specific methods, please refer to the test code provided by InternVL-Chat-V1-5 on HuggingFace (`pixel_values ​​= torch.stack(pixel_values)`). I think it is a more comprehensive way to test the "single-image input models" using both concat image and concat image token.

5. The author lacks some more advanced open-sourced models for evaluation, such as LLava—interleave [3], GLM-4V-9b [4], etc.


[1] V*: Guided Visual Search as a Core Mechanism in Multimodal LLMs
[2] How Far Are We to GPT-4V? Closing the Gap to Commercial Multimodal Models with Open-Source Suites
[3] LLaVA-NeXT-Interleave: Tackling Multi-image, Video, and 3D in Large Multimodal Models
[4] CogVLM: Visual Expert for Pretrained Language Models

**Questions:**

Please refer weaknesses for details.

---

> ### Author Response · Authors · 2024-11-24
>
> We appreciate the reviewer's insightful feedback. We provide a detailed response below to address the concerns and questions raised by the reviewer.
>
> > **W1. The first picture of Figure5 seems to be ambiguous**
>
> The confusion arises from the simplification made for presentation purposes. The [actual examples](https://bashify.io/i/sTeONM) (Please check the link pointing to an anonymous figure.) for this task include six to eight images rather than just three. We want to clarify that **the simplified conceptual example in the paper is to illustrate how we make the images, the question, and the options incompatible, in order to create an unanswerable instance**. One way to achieve this is by designing instances where the answer is non-deterministic in terms of the options (i.e., cannot be answered with certainty based on any given option so one should choose “none of the other options”).
>
> For **quality control**, as detailed in Section 3.2, we explain the process of **manually filtering** out errors and retaining only high-quality instances. Specifically, multiple experts reviewed all labeled instances to filter out uncertain cases. Additionally, we report **human performance**, which is higher than 93% accuracy, to further demonstrate the data quality.
>
> > **W2. Unanswerable setting experiment may result in "cheating"**
>
> This is a very insightful question. The pairwise design (answerable-unanswerable) of MuirBench is indeed intended to ensure that the model cannot achieve good performance by cheating. Cheating can arise from two sides:
> 1. Answering the answerable questions correctly by making an educated guess without fully understanding all the images.
> 2. Correctly identifying unanswerable questions due to the trend of overly refusing to answer.
>
> A reliable evaluation of robust multi-image understanding should avoid both problems. To address this, we ensure that each instance has a minimally revised counterpart with a different answer type (answerable <-> unanswerable).
> **Pairing each answerable instance with an unanswerable counterpart ensures robustness and reliability in evaluating** model performance.
>
> According to our analysis in Figure 8, the **current bottleneck** of advanced multimodal LLMs is related to **cheating on answerable questions** (type 1). The top five models achieve significantly better performance on the answerable set than on the unanswerable set. This highlights the challenges of ensuring robust understanding rather than leveraging shortcuts.
>
> Our benchmark provides a comprehensive testbed to analyze these cheating problems. We will include a discussion of this in the revised paper, incorporating findings from [1].
>
>
> > **W3. should the author’s definition of “Multi-Image Input Models” here be models trained with multi-image data?**
>
> Our original terminology refers to the intended use of the models. We will update the terms to avoid confusion based on the reviewer's suggestion to avoid confusion. We agree with the reviewer that it is possible to apply models trained with single-image data to conduct multi-image inference. To address this, we provide an **analysis in Appendix D investigating whether single-image-trained MLLMs can understand multi-image** input and vice versa. The results, presented in Table 3, indicate that models trained with single-image data can transfer their capability to process multi-image data to some extent. This also demonstrates the value of our benchmark in facilitating such analyses. We additionally present the results in the next section.
>
> > **W4. it is a more comprehensive way to test the "single-image input models" using both concat image and concat image token.**
>
> | Model | Input Format | Score |
> | - | - | - |
> | LLaVA-NeXT-34B | Concatenated Image | 33.31 |
> | LLaVA-NeXT-34B | Multiple Images (Concat image token) | 36.80 |
> | GPT-4o | Concatenated Image | 60.69 |
> | GPT-4o  | Multiple Images | 68.00 |
>
> > **W5. lacks some more advanced open-sourced models for evaluation, such as LLava—interleave ...**
>
> We report the results of most recent MLLMs, including **LLaVA-OneVision** (the subsequent version of LLava-Interleave; released in Aug. 2024) and Gemini-1.5-Pro (version 002 released in Sept. 2024).
>
> | Model | Overall Score |
> | - | -|
> | LLaVA-OneVision (0.5B) | 25.5 |
> | LLaVA-OneVision (7B) | 41.8 |
> | LLaVA-OneVision (72B) |  54.8 |
> | Gemini 1.5 Pro (001) | 64.9 |
> | Gemini 1.5 Pro (002) | 62.0 |

---

> ### Author Response · Authors · 2024-11-25
>
> Dear Reviewer QRSh,
>
> Thank you for your time and thoughtful feedback on our paper. We hope our responses have addressed your concerns, and we kindly request that you consider updating the score accordingly. If there are any remaining issues, please let us know, and we will be happy to provide further clarification.
>
> Thanks!

---

> ### Comment · Reviewer_QRSh · 2024-11-26
> **Reply to Authors**
>
> I appreciate the detailed response from the author, which addresses some of the potentially confusing points in the paper (such as the ambiguous images and definitions). By the way, I strongly recommend that the author highlight the changes in the updated PDF, as this would make the modifications clearer.
>
> However, based on the author's response, I still believe the paper has considerable room for improvement:
>
> - **About Manual Annotation:** The author repeatedly emphasizes that to indicate the quality of MuirBench in the rebuttal and the paper, but **the description of the manual annotation**  seems unclear. We only see a single figure showing the **interface** human annotations, but there is **no detailed information** about the number of annotators, their backgrounds, the time spent, etc.
>
> - **About the Unanswerable Setting:**
>
>   **First**, regarding the statement “Pairing each answerable instance with an unanswerable counterpart ensures robustness and reliability in evaluating model performance,” I do not see the importance of the unanswerable set design. **Why not simply randomize the answer choices and count as correct when the model answers both correctly?**  This seems to also achieve the purpose of enhancing the robustness of the evaluation. Isn’t this simpler and more direct? The unanswerable set should rather serve as an analysis tool, not as a key contribution.
>
>   **Secondly**, the conclusion in Figure 8 that “models cheat on answerable questions” seems overly absolute. Did the author try asking the models to provide explanations before making a choice? This could allow for manual checks to determine whether the models score higher on the answerable set due to random selection rather than not understanding the image.
>
>   **Thirdly**, I believe that for some models with poor performance on multi-image tasks, the conclusion drawn by the author may not hold, which can refer my first-round comments for further details.
>
> - **Concat Tokens for Evaluation:** The author adds LLaVA-Next-34B as a single-image model to test multi-image input, concluding that “explicitly processing multiple images as input is beneficial, as it reduces information loss and minimizes challenges in referencing images.” I acknowledge that this reduces information loss, but it introduces **the issue of long context input**, which is likely much longer than the context length seen during training (similar to LLMs). This also brings up potential **generalization issues**. Therefore, the conclusion drawn from testing just one model may be biased. I think it is necessary to test additional models to reach a more comprehensive conclusion.
>
> - **More Models:** Good work overall. I suggest adding to the paper, with some other new models such as InternVL2.
>
> - **Some Other Concerns:** After carefully reading the paper, I share the same concern as **[Reviewer 1gpk](https://openreview.net/forum?id=TrVYEZtSQH&noteId=zXpdZrRKLy)**. While I recognize that creating a stable benchmark for testing is a significant contribution, it is equally important to understand why certain models perform poorly and how improvements can be made (e.g., in architecture, data, or training strategies). This isn’t a request for the authors to train more models but rather to provide an analysis in this area. Although the author discusses this in Section 4.2, the focus is almost entirely on how to test. Therefore, I believe the paper lacks an analysis of how to improve multi-image capabilities, and this should be addressed.
>
> Given these reasons and other reviewer comments, I plan to maintain my score.

---

> ### Author Response · Authors · 2024-11-26
>
> We appreciate the reviewer’s active discussion and are happy to provide further clarification as follows.
>
> > **the description of the manual annotation**
>
> MuirBench was created by a team of more than 20 experts in this area, with annotations conducted by nine PhD students specializing in related fields. The annotated data underwent an additional review by three graduate-level students to ensure quality. We have provided the annotation details in **section 3.2 and appendix A.2**. If the reviewer has specific requests, we would be happy to provide further details.
>
> > **Why not simply randomize the answer choices**
>
> Randomizing answer choices **does not address the critical robustness concerns** introduced in this paper. There are more critical biases in addition to position bias in model evaluation, such as overfitting to specific patterns or failing to handle unanswerable questions effectively. Our approach deliberately includes unanswerable questions to assess the model's ability to abstain when appropriate, which is crucial for robust and reliable evaluation.
>
> > whether the models score higher on the answerable set due to random selection
>
> The answer choices are already randomly shuffled after data creation, ensuring fairness in evaluation.
>
> > models with poor performance on multi-image tasks
>
> The benchmark's goal is to identify potential issues and bottlenecks in models. For weaker models, the bottleneck may simply lie in their overall performance, which is still a critical insight for understanding their limitations. This does not diminish the value of our benchmark, as its primary purpose is to systematically identify model limitations and bottlenecks.
>
> > **Concat Tokens introduces the issue of long context input**
>
> The experiment was conducted **following the settings requested by the reviewer**. Definitely, there could be multiple causes for the model performance drop.
>
> Notably, a comprehensive analysis of specific model issues will be an independent work, separate from the scope of this general benchmark. The purpose of this (and any) benchmark is to provide a testbed for such analyses, and we already present initial results to encourage attention to these issues.
>
> > **InternVL2**
>
> In response to the reviewer's request, we evaluated InternVL2 on MuirBench. Notably, the results surpass some of larger models, such as LLaVA-v1.5-13B, highlighting the importance of the training process in addition to model size in achieving superior performance.
>
> | Model | Score |
> |-|-|
> | InternVL2-4B | 26.8 |
> | InternVL2-8B | 36.8 |
>
> We will definitely include the InternVL results in our paper. We are working on the paper update and will upload a new version with revised parts highlighted within one day.
>
> > **analysis of how improvements can be made (e.g., in architecture, data, or training strategies)**
>
> This paper, like any benchmark paper, focuses on identifying bottlenecks and providing a rigorous evaluation framework. The reviewer's request to propose ideas for improving model capabilities and provide supporting experiments is **beyond the scope of any benchmark paper**, as the purpose of such work is to assess and highlight limitations, not to propose solutions. Including this would be unreasonable within the constraints of a 10-page paper, especially given our already extensive 8-page appendix.

---

> > ### Comment · Reviewer_QRSh · 2024-12-02
> > **Response to author (Part2)**
> >
> > Also, It is also necessary to use the Concat Tokens method to conduct experiments, because the single-image model is also an important multimodal model, and the way to process multiple images is consistent with the multi-image model defined by the author. The author basically uses concat images for testing in the main text, which is actually an **unfair comparison** with the multi-image model.
> >
> > Although the author supplemented the concat tokens experiment in the rebuttal, it is difficult to draw a convincing conclusion through only one model. I think the concat tokens experiment should be used as the main test method for the single-image model, because this is consistent with the logic of multi-image model processing images.

---

> ### Comment · Reviewer_QRSh · 2024-11-27
> **Response to author**
>
> Thanks to the author for the detailed reply, I appreciate the author's efforts, but I still think there is large room for improvement in the article.
>
> **Concat Tokens introduces the issue of long context input**：Thank you for the author's question, but I don't think that "a comprehensive analysis of specific model issues will be an independent work". Given that what the author has done is a benchmark work, it is equally important to establish a complete evaluation system and a comprehensive analysis. There seems to be no essential difference between the multi-image model and the single-image model defined by the author except for the training data level, so I think this testing method needs to be discussed in detail.
>
> **InternVL2**: I just use InternVL2 as an example. I suggest the author to test some newer models, which will have greater reference value for the community.
>
> **beyond the scope of any benchmark paper** : **I am afraid I cannot agree with the author's such absolute description.** Simply presenting a dataset that makes the model perform poorly may not be a complete contribution. Of course, I recognize the quality of the MuirBench built by the author, but the current version of the paper obviously lacks some exploration of the solution, **which is also very important for benchmark papers**. Here are some examples:
>
> **M3COT(ACL Oral) Section 5.4: Exploration**
>
>
> **MMLU Section 5: Discussion**
>
>
> **MathVerse (ECCV 2024) Section 3.2 Experimental**

---

> > ### Author Response · Authors · 2024-11-27
> >
> > ### **Clarification of Our Claim**
> > We wish to clarify that we do not claim that benchmark papers do not require analysis. **Our claim is that the depth and scope of analysis in our paper are sufficient for a benchmark study.** We believe that the provided analysis is both insightful and impactful, offering valuable guidance for advancing MLLM development.
> >
> > ---
> >
> > ### **Addressing the Concern on Analysis Depth**
> > Since the primary concern revolves around the depth of analysis, we draw a direct comparison with the papers proposed by the reviewer. This comparison demonstrates that **our analysis is at least on par and, in some aspects, more comprehensive than the cited benchmarks, such as M3CoT and MathVerse.**
> >
> > ---
> >
> > ### **Analysis in MuirBench**
> > Below is a selected list of analyses and findings in this paper:
> > * In which multi-image tasks do multimodal LLMs show relative strengths and weaknesses?
> >   * We find that multi-image ordering and visual grounding appear to be more challenging for existing MLLMs, because these tasks require understanding the whole multi-image context and conducting more complicated reasoning processes across images and modalities afterwards. (lines 448-450)
> > * In which multi-image relations do multimodal LLMs show relative strengths and weaknesses?
> >   * We find that existing MLLMs fall short in understanding temporal relations and identifying cross-image similarity. (Table 2)
> > * Can models designed for single-image inputs perform multi-image tasks?
> >   * Our results show that models benefit from multi-image training data and learning processes to develop multi-image understanding capabilities. (lines 455-456)
> > * Do multimodal LLMs perform worse on the unanswerable set?
> >   * Our results reveals that models often fail to identify unanswerable questions, indicating that models may not comprehensively understand the multi-image context. (lines 463-465)
> > * Do image positions correlate with error rates?
> >   * Our results emphasize the necessity of training data covering diverse image positions to improve model generalizability. (lines 470-474)
> > * Do unanswerable types correlate with error rates?
> >   * This analysis demonstrate the distinct weaknesses of different MLLMs on this problem. (lines 477-480)
> > * Error analysis of GPT-4o
> >   * We identify the key weaknesses of GPT-4o in multi-image understanding, including capturing details, counting, and logical reasoning. (lines 485-500)
> > * Can multi-image MLLMs understand single-image input?
> >   * This analysis shows that concatenating images can lead to information loss and introduce coreference challenges between subimages and their textual mentions in the prompt. (lines 1065-1067)
> > * Can single-image MLLMs understand multi-image input?
> >   * The results suggest that explicitly processing multiple images as input is beneficial, as it reduces information loss and minimizes challenges in referencing images. (lines 1072-1074)
> >
> > ---
> >
> > ### **Comparison with M3CoT**
> >
> > Below, we outline our one-to-one comparisons to emphasize the scope of our analysis relative to M3CoT.
> >
> > - **M3CoT:** Multi-modal tool usage on text-modal LLMs fails.
> >   **MuirBench:** MLLMs trained mainly on single-image data perform poorly on multi-image understanding (Lines 452–456).
> >
> > - **M3CoT:** Performance cannot be boosted by text-only examples.
> >   **MuirBench:** Performance can drop when captioning images independently (Lines 1075–1079).
> >
> > - **M3CoT:** Performance may even be harmed by interleaved image and text examples.
> >   **MuirBench:** Error rate is correlated with image position (Lines 466–474).
> >
> > - **M3CoT:** Fine-tuning on M3CoT leads to better performance, and VLLMs benefit more than traditional VLMs.
> >   **MuirBench:** MLLMs trained with multi-image data perform significantly better (Lines 372–377).
> > ---

---

> ### Author Response · Authors · 2024-11-27
>
> ### **Comparison with MathVerse**
>
> Below, we outline our one-to-one comparisons to emphasize the scope of our analysis relative to MathVerse.
>
> - **MathVerse:** MLLMs rely more on diagram interpretation (DI) than seeing diagrams.
>   **MuirBench:** MLLMs may rely on shortcuts and fail to identify unanswerable instances (Lines 460–465).
>
> - **MathVerse:** MLLMs are moderately effective at perceiving implicit premises (IP).
>   **MuirBench:** MLLMs perform relatively better on image-text matching and visual retrieval (Lines 429–431).
>
> - **MathVerse:** MLLMs struggle to interpret explicit concepts (EC) from diagrams.
>   **MuirBench:** Tasks requiring understanding multi-image contexts and inter-image relations are more challenging (Lines 448–450).
>
> - **MathVerse:** MLLMs struggle to solve problems entirely using diagrams.
>   **MuirBench:** MLLMs struggle to fully comprehend multi-image contexts.  (Lines 372-377)
>
> - **MathVerse:** Closed-source MLLMs perform better.
>   **MuirBench:** Open-source models lag behind proprietary ones (Lines 374–375, Table 1).
>
> - **MathVerse:** LLMs achieve competitive results compared to MLLMs.
>   **MuirBench:** Multi-image understanding cannot be solved through dense captioning (Lines 1075–1079, Table 4).
>
> - **MathVerse:** GPT-4(V) outperforms humans on the text-only version.
>   **MuirBench:** Advanced MLLMs, such as GPT-4o, still fall short of satisfactory utility (Lines 372–373, Table 1).
>
> - **MathVerse:** Mathematical training improves performance.
>   **MuirBench:** Models trained with multi-image data achieves better performance (Lines 452–456, Table 1).
>
> ---
>
> ###  **Conclusion**
> The above comparisons demonstrate that the scope and depth of our analysis are comparable to those in MathVerse and M3CoT, both of which are regarded as impactful benchmarks. Additionally, we provide novel insights tailored to multi-image understanding, which distinguishes MuirBench from prior works. We hope this clarifies the value and comprehensiveness of our analysis.

---

> ### Author Response · Authors · 2024-11-27
>
> > **I just use InternVL2 as an example. I suggest the author to test some newer models, which will have greater reference value for the community.**
>
> We have already **tested nearly 30 recent MLLMs, including the models specifically requested by the reviewer, some of which are released within two months**. Additionally, we are committed to continually updating our leaderboard to include the latest models, ensuring the benchmark remains a valuable resource for the community.

---

> > ### Author Response · Authors · 2024-11-27
> >
> > Dear reviewer QRSh,
> >
> > To address your concerns regarding actionable insights, we have **added the following discussion on the opportunities for model improvement in Section 5** of the updated paper.
> >
> > ---
> >
> > **Opportunities for Model Improvement**
> >
> > Our findings highlight several opportunities for improving multimodal LLMs in multi-image scenarios. Multimodal LLMs struggle with tasks like multi-image ordering and visual grounding, which require complex reasoning across images and modalities, suggesting the need for more sophisticated training processes and model architectures that better integrate inter-image relationships. Additionally, models show weaknesses in understanding specific relations, such as temporal relation, which could be addressed by training on more temporally annotated data. Our results also reveal that models benefit from multi-image training. Thus, expanding multi-image datasets and training on diverse image types, tasks, and relations could improve generalization. Similarly, the model's performance drop on certain image positions suggests that training data should include a broader range of image positions. Furthermore, multimodal LLMs often fail to identify unanswerable questions, which are inevitable and common in the real world, pointing to the need for better training in recognizing insufficient context. Lastly, the challenge of inputting multiple images often requires compression or concatenation, which can lead to information loss or long-context issues. This highlights the need for new architectures that can process multiple images more effectively, preserving context and minimizing coreference challenges.

---

> ### Comment · Reviewer_QRSh · 2024-12-02
> **Response to author**
>
> Dear Author,
>
> Thank you for the explanation you provided. However, the comparison in the paper still does not fully address the limited insights provided by the article. First, the analysis presented still focuses mainly on the testing methods and simple conclusions drawn from the model performance analysis, such as "closed-source models perform better" or "multi-image training models perform better". **These conclusions seem somewhat predictable and may not provide substantial new insights to readers.** Since the effectiveness of closed-source models and expert models is widely accepted, I believe such a benchmark should provide more insightful conclusions.
>
> Second, **the paper currently** contains only 20 LVLMs, which is less than the "nearly 30 LVLMs" claimed. While I admit that 20 is still a good number, it is obviously not up to date, and reviewer 1gpk also raised similar concerns.
>
> I appreciate your efforts in the rebuttal stage, but like reviewer 1gpk, I believe that the current paper does not meet the standards for ICLR acceptance. Therefore, I decided to keep my score.
>
> Best regards

---

> ### Author Response · Authors · 2024-12-02
>
> > **These conclusions seem somewhat predictable and may not provide substantial new insights to readers.**
>
> We respectfully disagree with your assessment that the conclusions are predictable or lack substantial new insights. To the best of our knowledge, no prior work has provided the specific insights or empirical evidence presented in this paper. For example, we identify the weaknesses and strengths of multimodal LLMs on multi-image relations and tasks, which has not been addressed in any existing literature. Furthermore, the conclusions were derived through rigorous analysis of multimodal LLMs' multi-image understanding capabilities, which is novel and was explicitly designed to address the identified research gaps. These findings advance the understanding of multimodal LLMs and provide actionable implications for future research and real-world applications.
>
> Notably, the quoted "simple conclusion", "close-source models perform better," is from the MathVerse paper you recommended. We are presenting our analysis in the same format for your understanding.
>
> > **the paper currently contains only 20 LVLMs, which is less than the "nearly 30 LVLMs" claimed. it is obviously not up to date**
>
> Our original version included evaluations on 20 LVLMs. In response to your  request, we have expanded our experiments to include an additional 7 models, some of which were released just days around the ICLR submission deadline.
>
> > **concat images for testing is actually an unfair comparison. I think the concat tokens experiment should be used as the main test method**
>
> We respectfully disagree with your  assessment that the comparison is unfair. We follow the common practice in prior work of processing multi-image inputs by concatenating images for models trained on single-image data [1]. This aligns with the intended input format of these models.
>
> [1] Jiang, Dongfu, et al. "Mantis: Interleaved multi-image instruction tuning." arXiv preprint arXiv:2405.01483 (2024).

---

### Official Review · Reviewer_fMCn · 2024-11-04

**Soundness:** 3
**Presentation:** 3
**Contribution:** 3
**Rating:** 6
**Confidence:** 5

**Summary:**

This paper proposed a new benchmark aims at evaluating the performance of multi-modal language models for handling multiple image inputs.
The benchmark is designed following two key points, comprehensive coverage of tasks and robust evaluation.
For the comprehensive coverage of tasks, the paper seeks to leverage existings datasets from multiple sources as well as sourcing new data.
For the robust evaluation, the benchmark create unanswerable questions paired with each question.

Evaluation on the proposed benchmark demonstrates a gap of current multi-modal language models in understand multiple image inputs compared with human.

**Strengths:**

1. I think the paper is tackling an interesting and relativly novel setting, that is in evaluating the reasoning abilities of models across multiple inputs.
2. The experimental results demonstrate that there are indeed a gap for current models, and a possible direction for future research to be done.
3. The paper tested a range of state-of-the-art models.

**Weaknesses:**

1. The description of how the performance on unanswerable questions are measure is not very clear to me. Is "this question is unanswerable" part of the option or the evaluation just expect the model to give this?
2. Seems like the benchmark is unevenly distributed across different questions types and image types.
3. The benchmark consists of 10 different relations for multi-image, however, it is unclear to me how these 10 relations are chosen.

**Questions:**

1. I would like to know how the 10 relations are chosen, and how they can represent the use cases of multi-image understanding.
2. How is the performance on unanswerable questions measured.
3. The benchmark is imbalanced in terms of the question types and image types, how can one obtain one single score that can reflect the overall ability of the model to understand multiple images?
4. In L146, it is claimed that MIRB is released later than this paper. I'm wondering how is this claim supported? Given that this paper is still underreview.

---

> ### Author Response · Authors · 2024-11-24
>
> We appreciate the reviewer's insightful feedback. We provide a detailed response below to address the concerns and questions raised by the reviewer.
>
> > **W1/Q2. How is the performance on unanswerable questions measured.**
>
> “None of the other options” is included as **an option** for each instance. For unanswerable questions, we expect the model to select this option.
>
> > **W2/Q3.  the benchmark is unevenly distributed. how can one obtain one single score that can reflect the overall ability?**
>
> To facilitate fine-grained analysis of model performance, we have reported the performance on **each task** (Table 1) and **each multi-image relation** (Table 2) . This follows the common practice of many widely-used benchmarks, such as BLINK and MathVista.
>
> To obtain rebalanced results, one can also report the **task-balanced** or **relation-balanced** model performance (i.e., the macro average). We provide the reweighted performance below. While the numbers differ, the trends remain roughly the same.
>
>
> |                                                     | Avg   | Task-Macro Avg | Relation-Macro Avg |
> | --------------------------------------------------- | ----- | -------- | ------------ |
> | GPT-4o                                 | 68.00    | 58.74    | 65.63        |
> | GPT-4-Turbo                            | 62.31 | 56.24    | 60.85        |
> | Gemini Pro                  | 49.35 | 43.51    | 48.82        |
> | Mantis-8B-Idefics2          | 44.50  | 39.40    | 43.35        |
> | Mantis-8B-clip-llama3      | 37.38 | 34.14    | 36.99        |
> | Mantis-8B-siglip-llama3     | 36.12 | 33.02    | 37.62        |
> | Idefics-9B-Instruct | 35.43 | 32.35    | 28.00        |
> | Emu2-Chat (37B)          | 33.62 | 32.47    | 34.81        |
> | VILA1.5-13B                     | 33.12 | 30.46    | 33.37        |
> | Idefics2-8B                       | 26.08 | 26.65    | 35.46        |
> | OpenFlamingo-v2-9B | 23.73 | 23.90    | 23.87        |
> | LLaVA-NeXT-34B             | 33.31 | 32.21    | 34.28        |
> | LLaVA-v1.5-7B-xtuner            | 33.23 | 30.31    | 32.37        |
> | Yi-VL-6B                   | 28.69 | 28.40    | 30.24        |
> | LLaVA-internLM2-7B             | 28.15 | 27.26    | 28.95        |
> | LLaVA-v1.5-13B         | 24.38 | 23.92    | 23.63        |
> | LLaVA-v1.5-7B         | 23.46 | 23.25    | 24.47        |
> | LLaVA-v1.5-13B-xtuner            | 21.69 | 21.80    | 23.36        |
> | CogVLM                    | 20.85 | 21.47    | 23.17        |
>
> > **W3/Q1. how the 10 relations are chosen**
>
> We want to clarify that we did not decide or choose the multi-image relations to cover beforehand. Instead, we summarized the relations **based on prior studies and the data collected**, which **reflect the common focus of the research community**. During the development of MuirBench, we carefully reviewed the literature on multimodal models to collect multi-image data. The multi-image relations were then categorized in a bottom-up manner. Based on our expert manual observations, experience, and demonstrated examples, we believe our dataset is comprehensive enough to study common problems in existing multimodal LLMs.
>
> > **Q4. it is claimed that MIRB is released later than this paper**
>
> While the paper is still under review, our benchmark has been publicly available for some time to support the development of multimodal LLMs. Our benchmark is released before MIRB.

---

> > ### Comment · Reviewer_fMCn · 2024-11-27
> >
> > I would like to thank the author for the rebuttal.
> > The rebuttal has clear my concerns.
> > However, regarding Q4, I'm not sure if the author could claim that, give that this may break the double anonoymity. I believe it would be the best to delete this statement about release date from the paper and compare with MIRB in some other way.

---

> > > ### Author Response · Authors · 2024-12-04
> > >
> > > Dear Reviewer fMCn,
> > >
> > > As the discussion period is ending, we would like to thank you for volunteering your time in reviewing our paper and actively engaging in discussion. We appreciate your positive review of our paper and hope we have answered all your questions and addressed any concerns you had.

---

> ### Author Response · Authors · 2024-11-26
>
> Dear Reviewer fMCn,
>
> Thank you for your time and thoughtful feedback on our paper. We hope our responses have addressed your concerns, and we kindly request that you consider updating the score accordingly. If there are any remaining issues, please let us know, and we will be happy to provide further clarification.
>
> Thanks!

---

### Official Review · Reviewer_Yzea · 2024-11-04

**Soundness:** 3
**Presentation:** 4
**Contribution:** 3
**Rating:** 6
**Confidence:** 3

**Summary:**

This paper presents a new evaluation benchmark, dubbed MuirBench, which focuses on multi-image understanding capabilities of multimodal LLMs. Specifically, MuirBench contains 11,264 images and 2,600 multiple-choice questions, covering diverse multi-image relations. In experiments, the authors have evaluated 20 recent multi-modal LLMs, revealing that the best-performing models still struggled in multi-image understanding. Besides, the authors have also conducted detailed analyses, demonstrating possible reasons for the unsatisfying performance of multimodal LLMs.

**Strengths:**

- The evaluation aspect about multi-image understanding is novel and important for multimodal LLMs, which present significant contributions to the field.
- The proposed benchmark covers many types of multi-image relations, such as temporal, ordered-pages, or narrative relations, which are comprehensive to support solid evaluations for multimodal LLMs.
- The unanswerable counterpart of the benchmark presents a complementary perspective to assess multimodal LLMs fairly.

**Weaknesses:**

- It would be better if the authors could provide detailed comparison with recent multi-image evaluations benchmarks, such as Milebench, in a summarized table. This can help better capture the unique characteristics of MuirBench.
- The evaluation on MuirBench seems to require lots of computation resources, which may limit the accessibility for researchers with fewer resources.

**Questions:**

- Do the authors have plans to expand or adapt MuirBench to include new tasks or modalities as LVLM capabilities evolve?

---

> ### Author Response · Authors · 2024-11-24
>
> We appreciate the reviewer's insightful feedback. Below, we provide a detailed response to address the concerns and questions raised:
>
> > **W1. detailed comparison with recent multi-image evaluations benchmarks**
>
> Thanks for pointing out MileBench. We will include it in the related work section. Different from our work centering on robust multi-image understanding, MileBench primarily focuses on multimodal long-context scenarios. We will update the paper shortly.
>
> | Benchmark| #Test Instances | #Image/Instance | #Image Relation |  #Task |  #Image Domain |  Precise Metric | Unanswerable Questions |
> | -| -|-|-|-|-|-|-|
> | NLVR2 | 6967 | 2 | 1 | 1 | No Annotation |  Yes | No |
> | Blink | 1902 | 1-4 (majority 1 image) |  No Annotation | 9 (not all are multi-image tasks) | No Annotation | Yes | No |
> | MileBench-Realistic | 5197 | 2-109 (majority long context; avg 15 images + 422 words) | 2 | 10 | No Annotation | Partial | No |
> | Mementos | 699 | 2-25 | 1 | 1 | 3 | No | No |
> | Mantis-Eval | 217 | 2-5 | No Annotation | No Annotation | No Annotation | Yes | No |
> | MuirBench | 2600 | 2-9 (common context length; avg 4 Images + 38 words) | 10 | 12 | 12 |  Yes | Yes |
>
> > **W2. The evaluation on MuirBench seems to require lots of computation resources**
>
> We intentionally designed MuirBench to maintain a manageable data size, ensuring that computational resources remain reasonable. Most of the evaluations reported in the paper, including those using the GPT-4 API and LLaVA/Mantis on a single A100 GPU, take **less than half an hour** to complete.  This design choice ensures accessibility and practicality for researchers with varying computational budgets.
>
> > **Q1. Do the authors have plans to expand or adapt MuirBench to include new tasks or modalities as LVLM capabilities evolve?**
>
> Yes. We agree that as multimodal LLMs evolve, new challenges and requirements will emerge. We plan to continually expand MuirBench, adding more diverse scenarios and tasks to support the ongoing development and evaluation of multimodal LLMs. This commitment ensures that MuirBench remains a relevant and valuable benchmark for the community.

---

> > ### Comment · Reviewer_Yzea · 2024-11-26
> >
> > Thanks for your detailed response. My concerns are well solved. I will keep my rating.

---

> > > ### Author Response · Authors · 2024-12-04
> > >
> > > Dear Reviewer Yzea,
> > >
> > > As the discussion period is coming to a close, we would like to thank you for volunteering your time to review our paper. We sincerely appreciate your positive feedback and recognition of our contribution.

---

### Official Review · Reviewer_iBA5 · 2024-11-09

**Soundness:** 3
**Presentation:** 3
**Contribution:** 3
**Rating:** 6
**Confidence:** 2

**Summary:**

This paper introduces MUIRBENCH, a new benchmark designed to evaluate the capabilities of multimodal large language models (LLMs) in multi-image understanding. MUIRBENCH comprises 11,264 images and 2,600 multiple-choice questions spanning 12 multi-image tasks and 10 types of multi-image relations. Each task is paired with unanswerable counterparts to test robustness. Evaluations show that prominent models, like GPT-4 and Gemini Pro, perform far below human accuracy, emphasizing the need for improved models capable of holistic multi-image comprehension.

**Strengths:**

Let me first clarify that I am not an expert in evaluation dataset construction.
1) MUIRBENCH covers 12 different multi-image tasks and 10 image relationship categories to ensure the evaluation as comprehensive as possible.
2) Including answerable and unanswerable question variants can evaluate the robustness of models.
3) The image sources are expanded. In addition to existing multi-image data, medical, geographical, etc. are considered.
4) Human quality control and evaluation results are introduced to provide the evidence of faithfulness.

**Weaknesses:**

Let me first clarify that I am not an expert in evaluation dataset construction.

1) Lack of data scale comparison: I am not clear about the data scale of related multimodal evaluation benchmarks. It is recommended to provide a table to compare existing single-image/multi-image evaluation benchmarks, including the number of samples, the number of tasks, the number of relations, the number of image domain, etc. At least it is necessary to explain clearly how much improvement the proposed dataset in this paper has compared with previous work. I am not sure whether 1,300 answerable questions, 12 tasks and 10 relations are a relatively large improvement.

2) Specific explanation of comprehensiveness: I know that this paper considers 12 tasks and 10 relations, which sounds like a number that is neither too many nor too few. It is not clear whether there is evidence or analysis to state that this dataset is comprehensive enough at least within a certain range.

3) Are the ground-truth answers reliable? The human quality control does not seem to mention the analysis of the reliability of the answer. It seems that there is only human quality control of questions and images. I am not sure whether humans will find it difficult to judge or have no correct answers when answering these questions, and whether there will be subjective differences that lead to inconsistent evaluations by multiple people, which will add additional uncertainty to model evaluation.

**Questions:**

See Weaknesses.

---

> ### Author Response · Authors · 2024-11-24
>
> We appreciate the reviewer's insightful feedback. We provide a detailed response below to address the concerns and questions raised by the reviewer.
>
> > **W1. Lack of data scale comparison**
>
> The current data size is sufficient to advance the development of VLMs. For example, datasets like Winoground (CVPR 2022), which contains only 400 examples, have significantly influenced VLM advancements. Similarly, although focused on different aspects, other widely adopted benchmarks—including WHOOPS! (ICCV 2023), LlaVA-Bench (NeurIPS 2023), Visit-Bench (NeurIPS 2024), ConTextual (ICML 2024), VibeEval, and Visual Riddles (NeurIPS 2024)—comprise 90, 500, 576, 500, 269, and 400 examples, respectively, and have been pivotal for evaluating VLMs.
>
> We include comparisons with related benchmarks below. In summary, the size of our benchmark is **comparable to prior benchmarks**, and the total of 2,600 examples represents a comprehensive set. Importantly, the issue is not the size but the lack of a comprehensive benchmark for this critical problem: robust multi-image understanding. Our work provides the first comprehensive benchmark for evaluating multimodal LLMs' capabilities in this scenario.
>
>
> | Benchmark| #Test Instances | #Image/Instance | #Image Relation |  #Task |  #Image Domain |  Precise Metric | Unanswerable Questions |
> | -| -|-|-|-|-|-|-|
> | NLVR2 | 6967 | 2 | 1 | 1 | No Annotation |  Yes | No |
> | Blink | 1902 | 1-4 (majority 1 image) |  No Annotation | 9 (not all are multi-image tasks) | No Annotation | Yes | No |
> | MileBench-Realistic | 5197 | 2-109 (majority long context; avg 15 images + 422 words) | 2 | 10 | No Annotation | Partial | No |
> | Mementos | 699 | 2-25 | 1 | 1 | 3 | No | No |
> | Mantis-Eval | 217 | 2-5 | No Annotation | No Annotation | No Annotation | Yes | No |
> | MuirBench | 2600 | 2-9 (common context length; avg 4 Images + 38 words) | 10 | 12 | 12 |  Yes | Yes |
>
>
> > **W2. Specific explanation of comprehensiveness**
>
> During the development of MuirBench, we carefully examined over 30 related benchmarks and datasets to ensure broad coverage and relevance. The tasks and image relations in MuirBench are summarized based on prior studies and the data collected, **reflecting the common focus and priorities of the research community**. Guided by expert manual observations, extensive experience, and diverse demonstrated examples, we believe that MuirBench is comprehensive enough to address the most common challenges in existing multimodal models.
>
> That said, we acknowledge that certain tasks or relations may not yet be covered. We are committed to maintaining and expanding this project, incorporating new tasks and relations as the field evolves. Continuous updates will ensure that MuirBench remains a valuable and up-to-date resource for advancing robust multi-image understanding.
>
> > **W3. Are the ground-truth answers reliable?**
>
> The ground-truth **labels are reliable**. As detailed in Section 3.2, we implemented a rigorous manual filtering process to ensure the accuracy of the dataset. Multiple experts reviewed all labeled instances and filtered out those deemed uncertain, retaining only instances with correct answers. Notably, we reported **human performance** on the benchmark, which is over 93% accuracy, further demonstrating the quality of the annotations. Every instance in the benchmark includes a "no answer" option, allowing for the explicit handling of cases where the answer is genuinely uncertain. This design decision emphasizes data quality and supports fair evaluation of model capabilities.

---

> > ### Author Response · Authors · 2024-12-04
> >
> > Dear Reviewer iBA5,
> >
> > As the discussion period is ending, we would like to thank you for volunteering your time in reviewing our paper. We appreciate your positive review of our paper and hope we have answered all your questions and addressed any concerns you had.

---

### Author Response · Authors · 2024-11-29
**Summary of strengths recognized by the reviewers**

We appreciate the reviewers' insightful feedback. Below is a brief summary of strengths listed in their reviews:


- **Novelty and significance**: "novel and important, significant contributions to the field" (Yzea), "novel setting" (fMCn).


- **Comprehensive evaluation and insightful analysis**: "ensure the evaluation as comprehensive as possible" (iBA5), "comprehensive to support solid evaluations" (Yzea), "demonstrate gap and direction for future research" (fMCn), "comprehensive enough" (QRSh), "comprehensive benchmark, impressive analyses" (1gpk)


- **Robustness and Quality**: "provide the evidence of faithfulness" (iBA5), "assess multimodal LLMs fairly" (Yzea).

---

> ### Author Response · Authors · 2024-11-29
> **Summary of our responses and paper revision**
>
> We have provided detailed responses to address the concerns and questions raised by each reviewer. Below is a brief summary:
>
>
> - **Reviewer [iBA5](https://openreview.net/forum?id=TrVYEZtSQH&noteId=wSPhsFRJL8) and [Yzea](https://openreview.net/forum?id=TrVYEZtSQH&noteId=tpe1aMGnRz)**: The major request was to add a table comparing with prior benchmarks. We have added **Table 2**. Other clarification questions on computation cost and label reliability have been thoroughly addressed.
>
>
> - **Reviewer [fMCn](https://openreview.net/forum?id=TrVYEZtSQH&noteId=6sT1aXWZfz)**: The primary request was to add a balanced score. We have added it in **Table 4**. Other clarification questions on unanswerable options and the choice of relations have been thoroughly addressed.
>
>
> - **Reviewer [QRSh](https://openreview.net/forum?id=TrVYEZtSQH&noteId=fPaVhEEcF7) and [1gpk](https://openreview.net/forum?id=TrVYEZtSQH&noteId=zXpdZrRKLy)**:
>   - **Analysis scope**: Compared our analysis scope with the papers named by the reviewer in a [pairwise list](https://openreview.net/forum?id=TrVYEZtSQH&noteId=kJ8B7vBGPU).  Added **Section 5** to discuss the opportunities for model improvement.
>   - **More models**: We added 7 models requested by the reviewer (**Table 3**).
>   - Other minor points: We clarified the misunderstanding of Figure 5 and the unanswerable setting. We provided the balanced score in Table 4. The concat token analysis has already been presented in Table 6.  The request for inputting images one by one is already presented in Table 7.

---

### Meta-Review · Area_Chair_QUZt · 2024-12-22

**Metareview:**

(a) Scientific Claims and Findings

The paper introduces MuirBench, a benchmark designed to evaluate the multi-image understanding capabilities of multimodal large language models (LLMs). MuirBench includes 11,264 images and 2,600 multiple-choice questions across 12 tasks and 10 types of image relations, with unanswerable counterparts to test robustness. Evaluations reveal that current models, such as GPT-4 and Gemini Pro, perform significantly below human accuracy, highlighting the need for improved models. All reviewers note the benchmark's comprehensive coverage and its potential to reveal gaps in current model capabilities.

(b) Strengths

Reviewer iBA5 appreciates the comprehensive coverage of tasks and relations, as well as the inclusion of unanswerable questions to test robustness. Yzea highlights the novel focus on multi-image understanding and the benchmark's comprehensive nature. fMCn commends the benchmark's ability to reveal gaps in current models and its potential to guide future research. QRSh notes the detailed evaluation and discussion provided, while 1gpk emphasizes the benchmark's extensive coverage and the insights gained from testing recent VLMs.

(c) Weaknesses

The reviewers identify several weaknesses. iBA5 suggests providing a data scale comparison with existing benchmarks and questions the reliability of ground-truth answers. Yzea notes the lack of detailed comparison with recent benchmarks and the high computational resources required. fMCn finds the description of unanswerable question evaluation unclear and questions the choice of image relations. QRSh raises concerns about the ambiguity of unanswerable settings and the limitations of multi-image input models. 1gpk criticizes the lack of insightful analysis and novelty, the imbalance in question difficulty, and the limited practical application of unanswerable questions.

(d) Decision Reasons

On balance, AC agrees with positive points raised by reviewers which outweigh the negative points raised the two reviewers who favor rejection. The decision to accept the paper is based on its novel approach to evaluating multi-image understanding in multimodal LLMs and the comprehensive nature of the benchmark. The benchmark's ability to highlight gaps in current models and guide future research is a significant contribution, as noted by reviewers iBA5, Yzea, and fMCn. While there are concerns about the lack of detailed analysis and the practical application of unanswerable questions, the overall strengths in innovation, coverage, and potential impact on the field outweigh these weaknesses. The paper's contributions to advancing the evaluation of multimodal LLMs make it a valuable addition to the conference.

**Additional Comments On Reviewer Discussion:**

During the rebuttal period, the authors addressed several concerns raised by the reviewers, leading to a consensus for acceptance.
Reviewer iBA5 expressed that their concerns were addressed and decided to maintain their rating, indicating satisfaction with the authors' responses.

Reviewer Yzea also found their concerns well resolved and kept their rating, showing a positive reception to the rebuttal.
Reviewer fMCn appreciated the authors' rebuttal, which cleared their concerns. They suggested removing a statement about the release date to maintain double anonymity but maintained their rating, supporting acceptance.

Reviewer QRSh acknowledged the authors' detailed responses and recognized the quality of MuirBench, although they suggested areas for further improvement, such as testing newer models and providing a more comprehensive analysis. They appreciated the authors' efforts and the benchmark's potential value to the community.

Reviewer 1gpk appreciated the authors' comprehensive responses and the additional insights provided during the rebuttal. They acknowledged the practical aspects of the unanswerable questions and suggested further exploration of question formats and concatenation methods. Despite some remaining concerns, they recognized the benchmark's contribution to the field.

In the final decision, the authors' ability to address most reviewer concerns effectively during the rebuttal period was a significant factor. The positive feedback from reviewers iBA5, Yzea, and fMCn, along with the constructive engagement from QRSh and 1gpk, reinforced the decision to accept the paper. The contributions of MuirBench to advancing the evaluation of multimodal LLMs make it a valuable addition to the conference, with potential for further development and impact in future research.

The reviewers provided several points of constructive feedback for the authors to consider in the final version of the paper. Reviewer QRSh suggested simplifying the unanswerable set by exploring alternative setups, such as shuffling the option order, and emphasized the need for a more comprehensive and fair testing methodology, particularly for single-image models. They also noted that the analysis and discussion sections lacked depth and meaningful insights, and recommended addressing these areas to justify claims of comprehensive analysis, similar as reviewer 1gpk. Additionally, QRSh highlighted the importance of maintaining a constructive attitude during discussions and acknowledging all feedback, regardless of the reviewers' scores.

Reviewer Yzea agreed with the need for more in-depth analyses and clearer justification of the unanswerable set's necessity over other setups. They suggested focusing on distinctive insights from the evaluation results of MuirBench and cautioned against overemphasizing comparisons with previous studies.

Incorporating all reviewer suggestions will enhance the paper's contributions and impact, making it a more valuable resource for the community.

---

### Decision · Program_Chairs · 2025-01-22

Accept (Poster)